# Reciprocal Adversarial Learning via Characteristic Functions

**Shengxi Li**[*]    **Zeyang Yu**    **Min Xiang**    **Danilo Mandic**
Imperial College London
{shengxi.li17, z.yu17, m.xiang13, d.mandic}@imperial.ac.uk

## Abstract

Generative adversarial nets (GANs) have become a preferred tool for tasks involving complicated distributions. To stabilise the training and reduce the mode collapse of GANs, one of their main variants employs the integral probability metric (IPM) as the loss function. This provides extensive IPM-GANs with theoretical support for basically comparing moments in an embedded domain of the *critic*. We generalise this by comparing the distributions rather than their moments via a powerful tool, i.e., the characteristic function (CF), which uniquely and universally comprising all the information about a distribution. For rigour, we first establish the physical meaning of the phase and amplitude in CF, and show that this provides a feasible way of balancing the accuracy and diversity of generation. We then develop an efficient sampling strategy to calculate the CFs. Within this framework, we further prove an equivalence between the embedded and data domains when a reciprocal exists, where we naturally develop the GAN in an auto-encoder structure, in a way of comparing everything in the embedded space (a semantically meaningful manifold). This efficient structure uses only two modules, together with a simple training strategy, to achieve bi-directionally generating clear images, which is referred to as the reciprocal CF GAN (RCF-GAN). Experimental results demonstrate the superior performances of the proposed RCF-GAN in terms of both generation and reconstruction.

## 1   Introduction

Generative adversarial nets (GANs) owe their success to their powerful capability in capturing complicated data distributions [1]. In practical applications, however, their significant potential still remains under-explored as GANs typically suffer from unstable training and mode collapse issues [2]. An effective yet elegant way to address these issues is to replace the Jensen-Shannon (JS) divergence in measuring the discrepancy in the original form of GANs [3] by another class of metrics called the integral probability metric (IPM) [4] given by,

$$d(\mathcal{P}_d, \mathcal{P}_g) = \sup_{f \in \mathcal{F}} |\mathbb{E}_{x \sim \mathcal{P}_d}[f(x)] - \mathbb{E}_{x \sim \mathcal{P}_g}[f(x)]|, \tag{1}$$

where the symbol $\mathcal{F}$ in IPMs represents a collection of (typically real) bounded functions, $\mathcal{P}_g$ denotes the generated distribution, and $\mathcal{P}_d$ is the real data distribution. Using IPMs to improve GANs has been justified by the fact that in real-world data distributions are typically embedded in low-dimensional manifolds, which is intuitive because data preserve semantic information instead of being a collection of rather random pixels. Thus, the divergence measure ("bin-to-bin" comparison) of the original GAN could easily max out, whereas the IPMs such as the Wasserstein distance ("cross-bin" comparison) can consistently yield a meaningful measure between the generated and real data distributions [3].

---

[*]Corresponding author

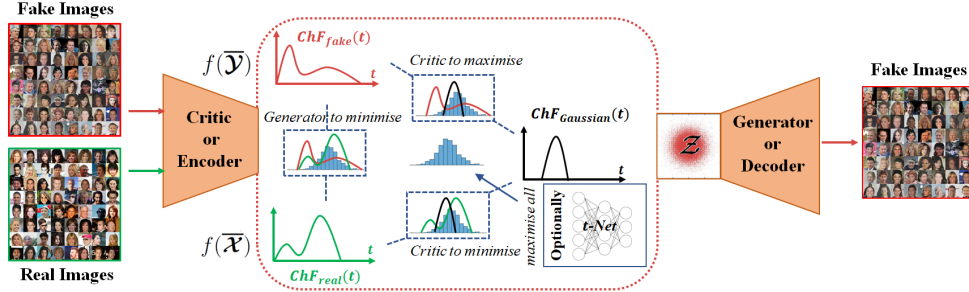

Figure 1: The overall structure of the proposed RCF-GAN. The generator serves to minimise the CF loss between the embedded real and fake distributions. The *critic* serves to minimise the CF loss between the embedded real and the input noise distributions, whilst maximising the CF loss between the embedded fake and the input noise distributions. Moreover, an MSE loss between the embedded fake and the input noise distributions is regularised as the auto-encoder loss, which has not been shown in the figure. An optional $t$-net can be employed to optimally sample the CF loss.

Varying collections of $\mathcal{F}$ in (1), therefore, defines different IPM-GANs and the supremum $\sup_{f \in \mathcal{F}}$ is then typically achieved by the discriminator net, or more formally, the *critic* in the IPM-GANs. The first IPM-GAN was motivated by the Wasserstein GAN (W-GAN) [5], where $\mathcal{F}$ denotes all the 1-Lipschitz functions. However, it has been widely argued that the *critic* is not powerful enough to search within all the 1-Lipschitz function spaces, which leads to limited diversity of the generator due to an ill-posed equivalence measurement of $\mathcal{P}_d$ and $\mathcal{P}_g$ [6, 7]. Follow-up works have been proposed to improve the W-GAN by either enhancing it to satisfy the 1-Lipschitz condition (e.g., by gradient penalty [8] or spectral normalization [9]) or by employing *easy-to-implement* $\mathcal{F}$ for the *critic*. The latter, by virtue of relaxing the *critic*, typically leads to a stringent comparison on the embedded feature domain, i.e., by matching higher-order moments instead of the mean matching in the W-GAN. This path includes many recent GANs which additionally consider the second-order moment (e.g., Fisher-GAN [10] and McGAN [11]), together with explicitly (e.g., Sphere GAN [12]) or implicitly (e.g., MMD-GAN [13, 14]) comparing higher-order moments. Furthermore, generalising (1) as moment matching problem has been justified as a natural and beneficial way to understand IPM-GANs [15–17]. This also compensates for the deficiency where the *critic* may not transform the data distributions into unimodal distributions, for example, the Gaussian distribution that is solely determined by the first- and second-order moments.

Moreover, it is more safe and elegant to compare the distributions because the equivalence in distributions ensures the equivalence in the moments; the inverse, however, does not necessarily hold. As a powerful tool of containing all the information relevant to a distribution, the *characteristic function* (CF) provides a universal way of comparing distributions, even when their probability density functions (pdfs) do not exist. The CF also has a one-to-one correspondence with the cumulative density function (cdf), which has also been verified to benefit the design of GANs [18]. Compared to the moment generating function (mgf) that has been reflected in the MMD-GAN [13], the CF is unique and universally existent. More importantly, the CF is automatically aligned at $\mathbf{0}$; this means that even a simple "bin-to-bin" comparison between CFs can consistently provide a meaningful measure and thus avoid gradient vanishing that appears in the original GAN [5]. On the other hand, the weak convergence property of CFs ensures that the convergence in the CF also indicates the convergence in the distributions.

In this paper, we propose a reciprocal CF GAN (RCF-GAN) as a natural generalisation of the existing IPM-GANs, with the overall structure shown in Fig. 1. It needs to be pointed out that incorporating the CF in a GAN is non-trivial because the CF is basically complex-valued and the comparison has to be performed on functions as well. To address these difficulties, we first demystify the role of CFs by finding that its phase is closely related to the distribution centre, whereas the amplitude dominates the distribution scale. This provides a feasible way of balancing the accuracy and diversity of generation. Then, as for the comparison over functions, we prove that other than in the whole space of CFs, sampling within a small ball around $0$ of CFs is sufficient to compare two distributions, and also enables the proposed CF loss to be bounded and differentiable almost everywhere. We further optimise the sampling strategy by automatically adjusting sampling distributions under the umbrella of the *scale mixture of normals* [19].

Benefiting from our powerful CF design in comparing distributions, we propose to purely compare in the embedded domain and prove its equivalence to the counterpart in the data domain when a reciprocal theory between the generator and the *critic* holds. This motivates us to incorporate an auto-encoder structure to satisfy this theoretical requirement. In this way, the *critic* in our RCF-GAN is further relaxed and only focuses on learning a fruitful embedding. Furthermore, different from many existing adversarial works with auto-encoders incorporating at least three modules[2] [13, 14, 21–26], our RCF-GAN only requires two modules that already exist in a GAN; the *critic* is an encoder and the generator is a decoder as well, which is neat and reasonable as this comes without increasing computational complexity and complicated (unstable) training strategies, as well as without other requirements such as the Lipschitz continuity. More importantly, the framework of comparing everything in the embedded domain enables the CF-GAN to learn a semantic and meaningful latent space, and to also avoid the smoothing artefact that arises from the use of point-wise mean square error (MSE) employed in the data domain. This benefits from both the auto-encoder and the GANs, i.e., *bi-directionally* generating *clear* images. Our experimental results show that our RCF-GAN achieves remarkable improvements on the generation, together with an additional capability in the reconstruction and interpolation[3].

## 2 Characteristic Function Loss and Efficient Sampling Strategy

### 2.1 Characteristic Function and Elliptical Distribution

The CF of a random variable, $\boldsymbol{\mathcal{X}} \in \mathbb{R}^m$, represents the expectation of its complex unitary transform, given by

$$\Phi_{\boldsymbol{\mathcal{X}}}(\mathbf{t}) = \mathbb{E}_{\boldsymbol{\mathcal{X}}}[e^{j\mathbf{t}^T\mathbf{x}}] = \int_{\mathbf{x}} e^{j\mathbf{t}^T\mathbf{x}} dF_{\boldsymbol{\mathcal{X}}}(\mathbf{x}), \tag{2}$$

where $F_{\boldsymbol{\mathcal{X}}}(x)$ is the cdf of $\boldsymbol{\mathcal{X}}$. We thus have $\Phi_{\boldsymbol{\mathcal{X}}}(\mathbf{0}) = 1$ and $|\Phi_{\boldsymbol{\mathcal{X}}}(\mathbf{t})| \leq 1$ for all $\mathbf{t}$. This property ensures that CFs can be straightforwardly compared in a "bin-to-bin" manner, because all CFs are automatically aligned at $\mathbf{t} = \mathbf{0}$. Moreover, when the pdf of $\boldsymbol{\mathcal{X}}$ exists, the expression in (2) is equal to its inverse Fourier transform; this ensures that $\Phi_{\boldsymbol{\mathcal{X}}}(\mathbf{t})$ is uniformly continuous. Another important property of the CF is that it uniquely and universally retains all the information regarding a random variable. In other words, a random variable does not necessarily need to possess a pdf (e.g., when it is an $\alpha$-stable distribution), but its CF always exists.

As the cdf, $F_{\boldsymbol{\mathcal{X}}}(\mathbf{x})$, is unknown and is to be compared, we employ the empirical characteristic function (ECF) as an asymptotic approximation in the form of $\widehat{\Phi}_{\boldsymbol{\mathcal{X}}_n}(\mathbf{t}) = \sum_{i=1}^{n} e^{j\mathbf{t}^T\mathbf{x}_i}$, where $\{\mathbf{x}_i\}_{i=1}^n$ are $n$ i.i.d. samples drawn from $\boldsymbol{\mathcal{X}}$. As a result of the *Levy continuity theorem* [28], the ECF converges weakly to the population CF [29]. More importantly, the *uniqueness theorem* guarantees that two random variables have the same distribution if and only if their CFs are identical [30]. Therefore, together with the weak convergence, the ECF provides a feasible and good proxy to the distribution, which has also been preliminarily applied in two sample test [31, 32]. Before proceeding further, we introduce an important class of distributions that will be used in this work.

**Example 1.** *Within unimodal distributions, one broad class of distributions is called the elliptical distribution, which is general enough to include various important distributions such as the Gaussian, Laplace, Cauchy, Student-t, $\alpha$-stable and logistic distributions. The elliptical distributions do not necessarily have pdfs, and we refer to [33] for more detail. The CF of an elliptical distribution, $\boldsymbol{\mathcal{X}}$, however, always exists and has the following form*

$$\Phi_{\boldsymbol{\mathcal{X}}}(\mathbf{t}) = e^{j\mathbf{t}^T\boldsymbol{\mu}}\psi(\mathbf{t}^T\boldsymbol{\Sigma}\mathbf{t}), \tag{3}$$

*where $\boldsymbol{\mu}$ denotes the distribution centre, $\boldsymbol{\Sigma}$ is the distribution scale, and $\psi(\cdot)$ is a real-valued function $\mathbb{R} \rightarrow \mathbb{R}$, for example, $\psi(s) = e^{(-s/2)}$ for the Gaussian distribution. By inspecting (3) we can see that the phase of the CF is solely related to the location of data centre and the amplitude is only governed by the distribution scale (diversity).*

## 2.2 Distance Measure via Characteristic Functions

The auto alignment property of the CFs allows us to incorporate a simple "bin-to-bin" comparison over two complex-valued CFs (corresponding to two random variables $\mathcal{X}$ and $\mathcal{Y}$), in the form

$$\mathcal{C}_{\mathcal{T}}(\mathcal{X}, \mathcal{Y}) = \int_{\mathbf{t}} \underbrace{\left((\Phi_{\mathcal{X}}(\mathbf{t}) - \Phi_{\mathcal{Y}}(\mathbf{t}))(\Phi_{\mathcal{X}}^*(\mathbf{t}) - \Phi_{\mathcal{Y}}^*(\mathbf{t}))\right)^{\frac{1}{2}}}_{c(\mathbf{t})} dF_{\mathcal{T}}(\mathbf{t}), \tag{4}$$

where $\Phi^*$ denotes the complex conjugate of $\Phi$ and $F_{\mathcal{T}}(\mathbf{t})$ is the cdf of a sampling distribution on $\mathbf{t}$. For the convenience of subsequent analysis, we represent the quadratic term for each $\mathbf{t}$ as $c(\mathbf{t}) = (\Phi_{\mathcal{X}}(\mathbf{t}) - \Phi_{\mathcal{Y}}(\mathbf{t}))(\Phi_{\mathcal{X}}^*(\mathbf{t}) - \Phi_{\mathcal{Y}}^*(\mathbf{t}))$. More importantly, $\mathcal{C}_{\mathcal{T}}(\mathcal{X}, \mathcal{Y})$ is a valid distance that measures the difference of two random variables via CFs, of which the proof is provided in Lemma 1; this means $\mathcal{C}_{\mathcal{T}}(\mathcal{X}, \mathcal{Y}) = 0$ if and only if $\mathcal{X} =^d \mathcal{Y}$. A specific type of $\mathcal{C}_{\mathcal{T}}(\mathcal{X}, \mathcal{Y})$ in (4) is when the pdf of $\mathbf{t}$ is proportional to $||\mathbf{t}||^{-1}$, and its relationship to other metrics, including the Wasserstein and Kolmogorov distances, has been analysed in detail [34].

**Lemma 1.** *The discrepancy between $\mathcal{X}$ and $\mathcal{Y}$, given by $\mathcal{C}_{\mathcal{T}}(\mathcal{X}, \mathcal{Y})$ in (4), is a distance metric when the support of $\mathcal{T}$ resides in $\mathbb{R}^m$.*

Furthermore, as the phase and amplitude of a CF indicate the data centre and diversity, we inspect $c(\mathbf{t})$ and rewrite it in a physically meaningful way, i.e., through the differences in the corresponding phase and amplitude terms as [35, 36],

$$\begin{aligned}
c(\mathbf{t}) &= |\Phi_{\mathcal{X}}(\mathbf{t})|^2 + |\Phi_{\mathcal{Y}}(\mathbf{t})|^2 - \Phi_{\mathcal{X}}(\mathbf{t})\Phi_{\mathcal{Y}}^*(\mathbf{t}) - \Phi_{\mathcal{Y}}(\mathbf{t})\Phi_{\mathcal{X}}^*(\mathbf{t}) \\
&= |\Phi_{\mathcal{X}}(\mathbf{t})|^2 + |\Phi_{\mathcal{Y}}(\mathbf{t})|^2 - |\Phi_{\mathcal{X}}(\mathbf{t})||\Phi_{\mathcal{Y}}(\mathbf{t})|(2\cos(\mathbf{a}_{\mathcal{X}}(\mathbf{t}) - \mathbf{a}_{\mathcal{Y}}(\mathbf{t}))) \\
&= |\Phi_{\mathcal{X}}(\mathbf{t})|^2 + |\Phi_{\mathcal{Y}}(\mathbf{t})|^2 - 2|\Phi_{\mathcal{X}}(\mathbf{t})||\Phi_{\mathcal{Y}}(\mathbf{t})| + 2|\Phi_{\mathcal{X}}(\mathbf{t})||\Phi_{\mathcal{Y}}(\mathbf{t})|(1 - \cos(\mathbf{a}_{\mathcal{X}}(\mathbf{t}) - \mathbf{a}_{\mathcal{Y}}(\mathbf{t}))) \\
&= \underbrace{(|\Phi_{\mathcal{X}}(\mathbf{t})| - |\Phi_{\mathcal{Y}}(\mathbf{t})|)^2}_{\text{amplitude difference}} + 2|\Phi_{\mathcal{X}}(\mathbf{t})||\Phi_{\mathcal{Y}}(\mathbf{t})| \underbrace{(1 - \cos(\mathbf{a}_{\mathcal{X}}(\mathbf{t}) - \mathbf{a}_{\mathcal{Y}}(\mathbf{t})))}_{\text{phase difference}},
\end{aligned} \tag{5}$$

where $\mathbf{a}_{\mathcal{X}}(\mathbf{t})$ and $\mathbf{a}_{\mathcal{Y}}(\mathbf{t})$ represent the angles (phases) of $\Phi_{\mathcal{X}}(\mathbf{t})$ and $\Phi_{\mathcal{Y}}(\mathbf{t})$, respectively. Therefore, we can clearly see that $\mathcal{C}_{\mathcal{T}}(\mathcal{X}, \mathcal{Y})$ basically measures the amplitude difference and the phase difference weighted by the amplitudes. We can further consider a convex combination of the two terms via $0 \leq \alpha \leq 1$, to yield

$$c_\alpha(\mathbf{t}) = \alpha\left((|\Phi_{\mathcal{X}}(\mathbf{t})| - |\Phi_{\mathcal{Y}}(\mathbf{t})|)^2\right) + (1-\alpha)\left(2|\Phi_{\mathcal{X}}(\mathbf{t})||\Phi_{\mathcal{Y}}(\mathbf{t})|(1 - \cos(\mathbf{a}_{\mathcal{X}}(\mathbf{t}) - \mathbf{a}_{\mathcal{Y}}(\mathbf{t})))\right). \tag{6}$$

Recall that for the elliptical distributions in Example 1, the phase represents the distribution centre while the amplitude represents the scale; $\mathcal{C}_{\mathcal{T}}(\mathcal{X}, \mathcal{Y})$ thus measures the both discrepancy of the centres and diversity of two distributions. We show in Figure 2-(a) that by swapping the phase and amplitude parts, the saliency information follows the phase part of the CF, which captures the centres of the distribution[4]. We further illustrate in Figure 2-(b) that this property still holds in real data distributions, even though they are much complicated and even non-unimodal. From Figure 2-(b)-(d), mainly training the phase (shown in Figure 2-(c)) results in generating images similar to an average of the real data, as a result of minimising the difference of the data centres. On the other hand, when mainly training the amplitude (shown in Figure 2-(b)), we can obtain diversified but inaccurate images ("wrong" numbers such as "1" for digit 7 and "6" for digit 5, uneven characters, disconnected artefacts, etc.). Therefore, by using different weights in $c_\alpha(\mathbf{t})$, we can flexibly capture the main content via minimising the phase difference, whilst enriching the diversity of generated images by increasing the amplitude loss. This provides a meaningful and feasible way of understanding the GAN loss in controlling the generation.

## 2.3 Sampling the Characteristic Function Loss

In practice, to calculate $\mathcal{C}_{\mathcal{T}}(\mathcal{X}, \mathcal{Y})$ efficiently, as mentioned in Section 2.1, $\Phi_{\mathcal{X}}(\mathbf{t})$ and $\Phi_{\mathcal{Y}}(\mathbf{t})$ can be evaluated by the ECFs of $\mathcal{X}$ and $\mathcal{Y}$, which are weakly convergent to the corresponding population CFs. The remaining task is to sample from $F_{\mathcal{T}}(\mathbf{t})$. A direct approach is to use the neural net where the input is Gaussian noise and the output is the samples of $F_{\mathcal{T}}(\mathbf{t})$. However, Proposition 1 indicates that this can lead to ill-posed optima whereby $F_{\mathcal{T}}(\mathbf{t})$ converges to some point mass distributions and

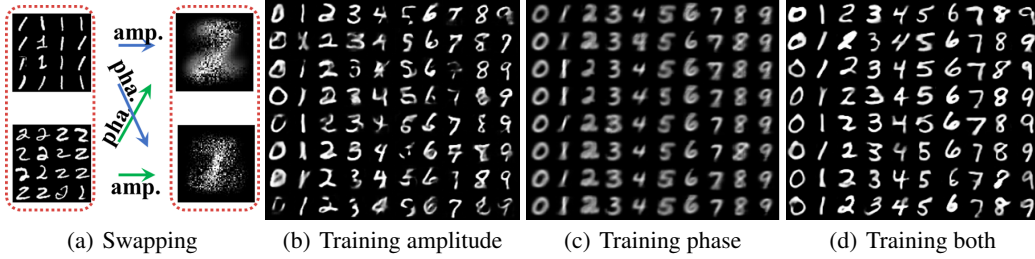

| (a) Swapping | (b) Training amplitude | (c) Training phase | (d) Training both |

Figure 2: Two experiments on the MNIST dataset which show the physical meaning of the phase and amplitude of the CF. (a) A multivariate Gaussian fit to the images of digits 1 and 2, by naively assuming that each pixel is independent from other pixels. Then, the phase and amplitude information of the CFs between the two multivariate distributions were swapped, and then randomly sampled from the swapped distributions. (b)-(d) A generator was directly trained on the given images of each digit. To avoid the impact from the *critic*, we *DO NOT* employ the *critic* in this experiment but directly calculate the loss between images after the generator with different $\alpha$. We performed training for amplitude for $\alpha = 0.999$ in (b), phase only for $\alpha = 0.001$ in (c) and equally training the amplitude and phase information for $\alpha = 0.5$ in (d).

thus is no longer supported in $\mathbb{R}^m$ as required in Lemma 1. In other words, for the degenerated $F_{\mathcal{T}}(\mathbf{t})$, we may have $\mathcal{C}_{\mathcal{T}}(\mathcal{X}, \mathcal{Y})$ but $\mathcal{X} \neq^d \mathcal{Y}$. In our experiment, we also found that directly optimising $F_{\mathcal{T}}(\mathbf{t})$ can cause instability.

**Proposition 1.** *The maximum of $\mathcal{C}_{\mathcal{T}}(\mathcal{X}, \mathcal{Y})$ is reached when $F_{\mathcal{T}}(\mathbf{t})$ attains a mass point at $\mathbf{t}^*$, where $\mathbf{t}^* = \arg\max_{\mathbf{t}} c(\mathbf{t})$. The minimum of $\mathcal{C}_{\mathcal{T}}(\mathcal{X}, \mathcal{Y})$ is reached when $F_{\mathcal{T}}(\mathbf{t})$ attains a mass point at $\mathbf{0}$.*

In the way of addressing this ill-posed optimisation on $F_{\mathcal{T}}(\mathbf{t})$, we can impose some constraints on $F_{\mathcal{T}}(\mathbf{t})$, for example, by assuming some parametric distributions. On the other hand, we may also be concerned that the constraints on $F_{\mathcal{T}}(\mathbf{t})$ can impede the ability of $\mathcal{C}_{\mathcal{T}}(\mathcal{X}, \mathcal{Y})$ as a metric to distinguish $\mathcal{X}$ from $\mathcal{Y}$. Lemma 2 provides an efficient and feasible way of choosing $F_{\mathcal{T}}(\mathbf{t})$.

**Lemma 2.** *If $\mathcal{X}$ and $\mathcal{Y}$ are supported on a finite interval $[-1, 1]^m$, $\mathcal{C}_{\mathcal{T}}(\mathcal{X}, \mathcal{Y})$ in (4) is still a distance metric for distinguishing $\mathcal{X}$ from $\mathcal{Y}$ for any $F_{\mathcal{T}}(\mathbf{t})$ that samples $\mathbf{t}$ within a small ball about $\mathbf{0}$.*

As shown in the next section, we employ $\mathcal{C}_{\mathcal{T}}(\mathcal{X}, \mathcal{Y})$ as the loss to compare two distributions from the *critic*. By employing bounded activation functions (tanh, sigmoid, etc.), the requirement of Lemma 2 is automatically satisfied. Therefore, instead of searching within all the real distribution spaces, the choices of $F_{\mathcal{T}}(\mathbf{t})$ can be safely restricted to some zero-mean distributions, e.g., the Gaussian distribution. Furthermore, compared to the fixed Gaussian distribution, it is preferable, whilst avoiding the ill-posed optimum, that $F_{\mathcal{T}}(\mathbf{t})$ could be optimised to better accommodate the difference between two distributions.

In this paper, we choose $F_{\mathcal{T}}(\mathbf{t})$ as the cdf of a broad class of distributions called the *scale mixture of normals*, in the form of

$$p_{\mathcal{T}}(\mathbf{t}) = \int_{\Sigma} p_{\mathcal{N}}(\mathbf{t}|\mathbf{0}, \Sigma) p_{\Sigma}(\Sigma) d\Sigma, \tag{7}$$

where $p_{\mathcal{T}}(\mathbf{t})$ is the pdf of $F_{\mathcal{T}}(\mathbf{t})$, while $p_{\mathcal{N}}(\mathbf{t}|\mathbf{0}, \Sigma)$ denotes the zero-mean Gaussian distribution with the covariance given by $\Sigma$, and $p_{\Sigma}(\Sigma)$ denotes distributions of $\Sigma$. It needs to be pointed out that the *scale mixture of normals* constitutes a large portion of the elliptical distributions and includes many important distributions (e.g., the Gaussian, Cauchy, Student-*t*, hyperbolic distributions [39]) by choosing different $p_{\Sigma}(\Sigma)$. Therefore, instead of directly optimising $F_{\mathcal{T}}(\mathbf{t})$, which leads to ill-posed solutions, we alternatively optimise the neural net to output the samples of $p_{\Sigma}(\Sigma)$. By using the affine transformation (or the re-parametrisation trick), we are able to propagate back the gradients.

We should point out that the term $\int_{\mathbf{t}} c(\mathbf{t}) dF_{\mathcal{T}}(\mathbf{t})$ contained in our CF loss can also be interpreted as certain well behaved kernels in the MMD metric. This is due to the fact that the shift invariant and characteristic kernels in the MMD metric have to satisfy $k(\mathbf{x}, \mathbf{y}) = \int_{\mathbf{t}} e^{-j\mathbf{t}^T(\mathbf{x}-\mathbf{y})} dF_{\mathcal{T}}(\mathbf{t})$ for some compactly supported $F_{\mathcal{T}}(\mathbf{t})$ [40]. In contrast to the predefined and fixed kernels in the MMD-GANs, the proposed optimisation on the types of $F_{\mathcal{T}}(\mathbf{t})$ is thus able to learn this important hyperparameter, i.e., the type of kernels. On the other hand, the elliptical distributions in Example 1 potentially provide a set of well-defined characteristic kernels, by choosing $F_{\mathcal{T}}(\mathbf{t})$ as a normalised version of the CFs in (3). Then, the corresponding real-valued kernels are the density generators in [19].

# 3 Reciprocal Adversarial Learning

## 3.1 Characteristic Function Loss in RCF-GAN

Although the CF loss is a complete metric for measuring any forms of data distributions (e.g., Fig. 2-(b)-(d)), the CF loss in (4) works more efficiently and effectively in the embedded domain, with higher likelihood of learning fruitful representations of data. To this end, we first express our RCF-GAN in the IPM-GAN format as

$$d(\mathcal{P}_d, \mathcal{P}_g) = \sup_{\mathcal{T}, f \in \mathcal{F}} \mathcal{C}_{\mathcal{T}}(f(\overline{\mathcal{X}}), f(\overline{\mathcal{Y}})), \ \ \overline{\mathcal{X}} \sim \mathcal{P}_d \text{ and } \overline{\mathcal{Y}} \sim \mathcal{P}_g, \tag{8}$$

where we make a distinction between the random variables ($\overline{\mathcal{X}}$ and $\overline{\mathcal{Y}}$) in the data domain and those ($\mathcal{X}$ and $\mathcal{Y}$) in the embedded domain, i.e., $\mathcal{X} =^d f(\overline{\mathcal{X}})$ and $\mathcal{Y} =^d f(\overline{\mathcal{Y}})$. Lemma 3 below shows that this metric is well-defined for neural net training.

**Lemma 3.** *The metric $\mathcal{C}_{\mathcal{T}}(\mathcal{X}, \mathcal{Y})$ is bounded and differentiable almost everywhere.*

Because $\mathcal{C}_{\mathcal{T}}(\cdot, \cdot)$ is bounded by construction, it relaxes the requirements on the *critic* $f \in \mathcal{F}$. Otherwise, we may need to bound $\mathcal{F}$ to ensure the existence of the supremum [10].

## 3.2 Matching in the Embedded Space

Having proved that $\mathcal{C}_{\mathcal{T}}(\mathcal{X}, \mathcal{Y}) = 0 \Leftrightarrow \mathcal{X} =^d \mathcal{Y}$, we also need to prove the equivalence between $\mathcal{C}_{\mathcal{T}}(f(\overline{\mathcal{X}}), f(\overline{\mathcal{Y}})) = 0$ and $\overline{\mathcal{X}} =^d \overline{\mathcal{Y}}$, to ensure that our RCF-GAN correctly learns the real distribution in the data domain. This result is provided in Lemma 4.

**Lemma 4.** *Denote the distribution mapping by $\overline{\mathcal{Y}} =^d g(\mathcal{Z})$. Given two functions $f(\cdot)$ and $g(\cdot)$ that map between the supports of $\overline{\mathcal{Y}}$ and $\mathcal{Z}$, if $\mathbb{E}_{\mathcal{Z}}[||\mathbf{z} - f(g(\mathbf{z}))||_2^2] = 0$, we also have the reciprocal property $\mathbb{E}_{\overline{\mathcal{Y}}}[||\overline{\mathbf{y}} - g(f(\overline{\mathbf{y}}))||_2^2] = 0$, and vice versa. More importantly, this yields the following equivalences: $\mathcal{C}_{\mathcal{T}}(f(\overline{\mathcal{X}}), f(\overline{\mathcal{Y}})) = 0 \Leftrightarrow \mathcal{C}_{\mathcal{T}}(\overline{\mathcal{X}}, \overline{\mathcal{Y}}) = 0 \Leftrightarrow \mathcal{C}_{\mathcal{T}}(f(\overline{\mathcal{Y}}), \mathcal{Z}) = 0 \text{ and } \mathcal{C}_{\mathcal{T}}(f(\overline{\mathcal{X}}), \mathcal{Z}) = 0.$*

As a prerequisite of Lemma 4, the co-domains between $f(\cdot)$ and $g(\cdot)$ need to reside on the supports of $\overline{\mathcal{Y}}$ and $\mathcal{Z}$. Otherwise, the reciprocal may not hold. In our RCF-GAN, we propose an anchor design to our *critic*, by rewriting the *critic* loss (by minimising) as $-(C_{\mathcal{T}}(f(\overline{\mathcal{Y}}), \mathcal{Z}) - C_{\mathcal{T}}(f(\overline{\mathcal{X}}), \mathcal{Z}))$. Thus, $\mathcal{Z}$ operates as the static anchor (or pivot) in the dynamic training process. Besides stabilising and improving the convergence in training, this further enables the *critic* to quickly map real data, $\overline{\mathcal{X}}$, to the support of $\mathcal{Z}$, whilst the generator tries to map the generated distribution, $\overline{\mathcal{Y}}$, to the real data, $\overline{\mathcal{X}}$. The adversarial part to maximise $C_{\mathcal{T}}(f(\overline{\mathcal{Y}}), \mathcal{Z})$ aims to improve the generation quality against the generator loss, i.e., $\mathcal{C}_{\mathcal{T}}(f(\overline{\mathcal{X}}), f(\overline{\mathcal{Y}}))$. Fig. 1 illustrates the triangle relationship in our anchor design.

Furthermore, Lemma 4 indicates that instead of being regarded as components of some IPMs (e.g., the W-GAN) to be optimised with strict restrictions, the *critic* can be basically regarded as a feature mapping because in the embedded domain the CF loss is a valid distance metric of distributions. The *critic* can then be relaxed to satisfy the reciprocal property. Therefore, we incorporate the auto-encoder in only two modules by interchangeably treating the *critic* as the encoder and the generator as the decoder. More importantly, Lemma 4 ensures that matching in the embedded space is sufficient due to $\mathbb{E}_{\mathcal{Z}}[||\mathbf{z} - f(g(\mathbf{z}))||_2^2] = 0 \rightarrow \mathbb{E}_{\overline{\mathcal{Y}}}[||\overline{\mathbf{y}} - g(f(\overline{\mathbf{y}}))||_2^2] = 0$. This is beneficial in various applications such as the image generation (and reconstruction), where in the data domain, the MSE loss typically leads to smooth artefacts.

## 3.3 Putting Everything Together

In practice, in Lemma 4, we regard $f(\cdot)$ as the *critic* and $g(\cdot)$ as the generator. The $t$-net is denoted by $h(\cdot)$ and the covariance matrix of its output is assumed to be diagonal (we thus represent it as $\boldsymbol{\sigma}$), which is reasonable as in the embedded domain the multiple dimensions tend to be uncorrelated [41]. We also need to clarify that because the $t$-net is optional and in our RCF-GAN, fixed Gaussian can be directly sampled for $\mathbf{t}$, we separate the $t$-net from $f(\cdot)$. However, if the $t$-net is employed, since they (the $t$-net and *critic*) have the same goal of distinguishing the generated distribution from the real data distribution, they are optimised simultaneously and share the same *critic* loss, i.e., $-(C_{\mathcal{T}}(f(\overline{\mathcal{Y}}), \mathcal{Z}) - C_{\mathcal{T}}(f(\overline{\mathcal{X}}), \mathcal{Z}))$. Moreover, the *critic* additionally minimises an MSE loss to ensure the reciprocal property. On the other hand, the generator is trained by minimising (8) as usual. The pseudo-code for the proposed RCF-GAN is provided in Algorithm 1.

It also needs to be pointed out that here we choose $\mathcal{Z}$ as the Gaussian distribution for a fair comparison to other GANs; other complex distributions can be seamlessly adopted in our framework

according to different tasks, for example, finite mixture models for un-supervised and semi-supervised classifications, and learnt distributions for sequential data processing.

**Remark 1.** *Besides the case of computation, the structure of the proposed RCF-GAN benefits from its interpretation as both a GAN and an auto-encoder, as a way of unifying them. As an auto-encoder, the RCF-GAN enables us to compare reconstructions solely on a meaningful embedded manifold, instead of in the data domain. When regarded as a GAN, the auto-encoder part theoretically and practically indicates the convergence; it also stabilises the training by pushing the embedded distributions to the static anchor $\mathcal{Z}$.*

---

**Algorithm 1:** RCF-GAN. In all the experiments in this paper, the generator and the *critic* are trained once at each iteration. The optional *t*-net with parameter $\boldsymbol{\theta}_t$ is designated by $h_{\boldsymbol{\theta}_t}(\cdot)$.

---

**input:** Real data distribution $\mathcal{P}_d$; Gaussian noise $\mathcal{P}_{\mathcal{N}}$; batch sizes $b_d$, $b_g$, $b_t$ and $b_\sigma$ for the data, the generator input noise, $\mathcal{T}$ and *t*-net input noise, respectively; learning rate $l_r$; reciprocal regularisation in the embedded domain $\lambda$

**output:** Net parameters $\boldsymbol{\theta}_c$ and $\boldsymbol{\theta}_g$ for the *critic* and generator, respectively

**while** $\boldsymbol{\theta}_c$ *and* $\boldsymbol{\theta}_g$ *not converge* **do**

    `/* train the critic */`

    Sample from distributions: $\{\overline{\mathbf{x}}_i\}_{i=1}^{b_d} \sim \mathcal{P}_d$; $\{\mathbf{z}_i\}_{i=1}^{b_g} \sim \mathcal{P}_{\mathcal{N}}$; $\{\mathbf{t}_i\}_{i=1}^{b_t} \sim \mathcal{P}_{\mathcal{N}}$; $\{\boldsymbol{\sigma}_i\}_{i=1}^{b_\sigma} \sim \mathcal{P}_{\mathcal{N}}$

    Affine transform: $\{\mathbf{t}_i\}_{i=1}^{b_t} \leftarrow \big(\{\mathbf{t}_i\}_{i=1}^{b_t}, h_{\boldsymbol{\theta}_t}(\{\boldsymbol{\sigma}_i\}_{i=1}^{b_\sigma})\big)$          `// optional`

    Calculate adversarial loss:     `// emperical version of` $-\big(\mathcal{C}_{\mathcal{T}}(f(\overline{\boldsymbol{\mathcal{Y}}}), \boldsymbol{\mathcal{Z}}) - \mathcal{C}_{\mathcal{T}}(f(\overline{\boldsymbol{\mathcal{X}}}), \boldsymbol{\mathcal{Z}})\big)$

$$\mathcal{L} = -\big(\mathcal{C}_{\{\mathbf{t}_i\}_{i=1}^{b_t}}\big(f_{\boldsymbol{\theta}_c}(g_{\boldsymbol{\theta}_g}(\{\mathbf{z}_i\}_{i=1}^{b_g})), \{\mathbf{z}_i\}_{i=1}^{b_g}\big) - \mathcal{C}_{\{\mathbf{t}_i\}_{i=1}^{b_t}}\big(f_{\boldsymbol{\theta}_c}(\{\overline{\mathbf{x}}_i\}_{i=1}^{b_d}), \{\mathbf{z}_i\}_{i=1}^{b_g}\big)\big)$$

    Update: $\boldsymbol{\theta}_t \leftarrow \boldsymbol{\theta}_t + l_r \cdot \mathrm{Adam}(\boldsymbol{\theta}_t, \nabla_{\boldsymbol{\theta}_t}[\mathcal{L}])$

$$\boldsymbol{\theta}_c \leftarrow \boldsymbol{\theta}_c + l_r \cdot \mathrm{Adam}\big(\boldsymbol{\theta}_c, \nabla_{\boldsymbol{\theta}_c}[\mathcal{L} + \lambda \textstyle\sum_{i=1}^{b_g} ||\mathbf{z}_i - f_{\boldsymbol{\theta}_c}(g_{\boldsymbol{\theta}_g}(\mathbf{z}_i))||_2^2]\big)$$

    `/* train the generator */`

    Sample from distributions: $\{\overline{\mathbf{x}}_i\}_{i=1}^{b_d} \sim \mathcal{P}_d$; $\{\mathbf{z}_i\}_{i=1}^{b_g} \sim \mathcal{P}_{\mathcal{N}}$; $\{\mathbf{t}_i\}_{i=1}^{b_t} \sim \mathcal{P}_{\mathcal{N}}$; $\{\boldsymbol{\sigma}_i\}_{i=1}^{b_\sigma} \sim \mathcal{P}_{\mathcal{N}}$

    Affine transform: $\{\mathbf{t}_i\}_{i=1}^{b_t} \leftarrow \big(\{\mathbf{t}_i\}_{i=1}^{b_t}, h_{\boldsymbol{\theta}_t}(\{\boldsymbol{\sigma}_i\}_{i=1}^{b_\sigma})\big)$          `// optional`

    Calculate adversarial loss:            `// emperical version of` $\mathcal{C}_{\mathcal{T}}(f(\overline{\boldsymbol{\mathcal{Y}}}), f(\overline{\boldsymbol{\mathcal{X}}}))$

$$\mathcal{L} = \mathcal{C}_{\{\mathbf{t}_i\}_{i=1}^{b_t}}\big(f_{\boldsymbol{\theta}_c}(g_{\boldsymbol{\theta}_g}(\{\mathbf{z}_i\}_{i=1}^{b_g})), f_{\boldsymbol{\theta}_c}(\{\overline{\mathbf{x}}_i\}_{i=1}^{b_d})\big)$$

    Update: $\boldsymbol{\theta}_g \leftarrow \boldsymbol{\theta}_g + l_r \cdot \mathrm{Adam}(\boldsymbol{\theta}_g, \nabla_{\boldsymbol{\theta}_g}[\mathcal{L}])$

---

# 4 Experimental Results

In this section, our RCF-GAN is evaluated in terms of both image generation, reconstruction and interpolation, with our code available at `https://github.com/ShengxiLi/rcf_gan`. We also show in the supplementary material advanced results including phase and amplitude analysis, ablation study and superior performances under the ResNet structure.

**Datasets:** Three widely applied benchmark datasets were employed in the evaluation: CelebA (faces of celebrities) [44], CIFAR-10 [45] and LSUN Bedroom (LSUN_B) [46]. The images of the CelebA and LSUN_B were cropped to the size $64 \times 64$, whist the image size of the CIFAR10 was $32 \times 32$. When evaluating the reconstruction, the test sets of the CIFAR10 and LSUN_B were employed, of which the samples were not used in the training.

**Baselines:** As our work is mainly related to the IPM-GANs, we compared our RCF-GAN with the W-GAN [5], W-GAN with gradient penalty (W-GAN-GP) [8] and MMD-GAN [13, 14]. As an advancement of the MMD-GAN, the most recent work, OCF-GAN [27], together with its gradient penalty version (OCF-GAN-GP) was also compared. We need to point out that all the results reported in [27] were evaluated for the image size of $32 \times 32$. We thus ran the experiments for the CelebA and LSUN_B for image sizes $64 \times 64$ by using its provided code. For image reconstruction, we compared our RCF-GAN with the recent adversarial generator-encoder (AGE) work [20], which empirically performs better than the adversarially learned inference (ALI) [26].

**Metrics:** The Fréchet inception distance (FID) [43] was employed as a performance metric, which is basically the Wasserstein distance between two Gaussian distributions, together with the kernel inception distance (KID) that arises from the MMD metric [14]. In evaluating the FID and KID scores, we randomly generated 25,000 samples for both generation and true images, and obtained these metrics in terms of mean and standard deviation by 10 times repeated random selections.

Table 1: The FID and KID scores obtained from the DCGAN [42] structure. The results of the DCGAN and W-GAN-GP are from [43] and [14]. The corresponding publicly available codes were run to obtain the results of the W-GAN [5], MMD-GAN [13], OCF-GAN and OCF-GAN-GP [27]. The results of the AGE were tested from its pre-trained models [20].

| Methods | FID | | | KID | | |
|---|---|---|---|---|---|---|
| | CIFAR-10 | Celeba | LSUN_B | CIFAR-10 | Celeba | LSUN_B |
| DCGAN | 37.7 [43] | 21.4 [43] | 70.4 [43] | —- | —- | —- |
| W-GAN | 42.64±0.26 | 31.85±0.28 | 57.05±0.37 | 0.025±0.001 | 0.023±0.001 | 0.048±0.002 |
| W-GAN-GP | 37.52±0.19[14] | —- | 41.39±0.25[14] | 0.026±0.001[14] | —- | 0.039±0.002[14] |
| MMD-GAN | 42.8±0.27 | 32.5±0.16 | 56.52±0.34 | 0.025±0.001 | 0.024±0.001 | 0.047±0.002 |
| OCF-GAN | 40.99±0.15 | 32.66±0.16 | 61.48±0.23 | 0.024±0.001 | 0.024±0.001 | 0.052±0.002 |
| OCF-GAN-GP | 33.68±0.21 | 16.09±0.25 | 65.18±0.317 | 0.021±0.001 | 0.011±0.001 | 0.060±0.002 |
| AGE | 32.54±0.24 | 23.19±0.14 | —- | 0.020±0.001 | 0.017±0.001 | —- |
| RCF-GAN$_{(t\_norm)}$ | 31.55±0.20 | 19.34±0.22 | **38.16±0.286** | 0.019±0.001 | 0.012±0.001 | **0.032±0.001** |
| RCF-GAN$_{(t\_net)}$ | **31.21±0.21** | **15.86±0.08** | 40.15±0.40 | **0.018±0.001** | **0.011±0.001** | 0.034±0.001 |
| AGE(R) | 47.37±0.32 | 30.77±0.19 | —- | 0.022±0.001 | 0.024±0.001 | —- |
| RCF-GAN$_{(t\_net)}$(R) | **28.70±0.16** | **14.82±0.12** | 44.16±0.42 | **0.014±0.001** | **0.009±0.000** | 0.036±0.001 |

Note: $t\_norm$ corresponds to use the fixed Gaussian samples and $t\_net$ to the t-net. (R) denotes for the reconstruction.

**Net structure and technical details:** For a fair comparison, all the reported results were compared under the batch sizes of 64 (i.e., $b_d = b_g = b_t = b_\sigma = 64$). Moreover, all variances of Gaussian noise were set to 1, except for the input noise of the generator that was 0.3, because the reciprocal loss had to be minimised given the fact that the output of the *critic* is restricted to $[-1, 1]$. Furthermore, we do not require the Lipschitz constraint, which allows for a relatively larger learning rate ($l_r = 0.0002$ for both nets). Moreover, for the CIFAR10 and LSUN_B datasets, the dimension of the embedded domain was set to 128 and for the CelebA dataset the dimension was 64. The optional *t*-net, if used, was a small three layer fully connected net, with the dimension of each layer being the same as the embedded dimension. Our default RCF-GAN used *t*-net and layer normalisation, and was trained with the vanilla CF loss (i.e., $\alpha = 0.5$ in (6)).

**Image generation:** The images generated from random Gaussian noise are shown in Fig. 3. Observe that by using the proposed CF loss in the RCF-GAN, the generated images are clear and close to the real images; the FID and KID scores are further provided in Table 1. This table shows that the proposed RCF-GAN consistently achieved the best performances across the three datasets. The OCF-GAN-GP achieved comparable generation performance on the CelebA dataset, but had relatively inferior performances compared to our RCF-GAN on the CIFAR-10 and LSUN_B datasets. Thus, although the most recent independent work, OCF-GAN, also adopts the characteristic function in designing the loss, it still operates under the MMD-GAN framework, without the interpretation of the physical meaning of the characteristic function and the consideration of the *t*-net proposed in this paper. More importantly, the reciprocal structure introduced in this paper, together with the proposed CF loss, stably and significantly improves the image generation performance.

By inspecting the achieved best performances of RCF-GAN, the use of the *t*-net in outputting optimal $F_{\mathcal{T}}(\mathbf{t})$ proved beneficial. Moreover, solely training $g(\mathbf{z})$ via the CF typically performs inferior, which in our experiments on CelebA, obtained a 165 FID score (i.e., rough faces). This also verifies the benefit of latent space comparison via our critic. We also need to point out that in the default setting, our *critic* and generator were evaluated under almost the same number of model parameters as W-GANs, whereas MMD-GANs need an extra decoder net. The only extra cost in our *t*-net is negligible because it is a 3-layer fully connected net with the dimension of each layer less than 128.

More importantly, compared to a fluctuated generator loss that is caused by the adversarial module in GANs, we take the advantages of the auto-encoder structure in utilising the reciprocal loss (i.e., $\mathbb{E}_{\mathcal{Z}}[||\mathbf{z} - f(g(\mathbf{z}))||_2^2]$ indicates the reciprocal loss in the embedded space), together with the distance between the embedded real distribution $f(\overline{\mathcal{X}})$ and the Gaussian distribution $\mathcal{Z}$ (i.e., $\mathcal{C}_{\mathcal{T}}(f(\overline{\mathcal{X}}), \mathcal{Z})$) to better indicate the convergence, as shown in Figure 3. Intuitively, the reciprocal loss measures the convergence on reconstructions, whereas the real image embedding distance $\mathcal{C}_{\mathcal{T}}(f(\overline{\mathcal{X}}), \mathcal{Z})$ indicates the performance on generating images.

**Image reconstruction:** Benefiting from the reciprocal requirement introduced in Lemma 4, the proposed RCF-GAN can also reconstruct images and learn a semantic meaningful space. Images reconstructed and interpolated by RCF-GAN, AGE and MMD-GAN are shown in Fig. 4. As seen from this figure, because the RCF-GAN only matches the distributions in the embedded domain, the reconstructed images are thus clear and semantically meaningful, resulting in a superior interpolation

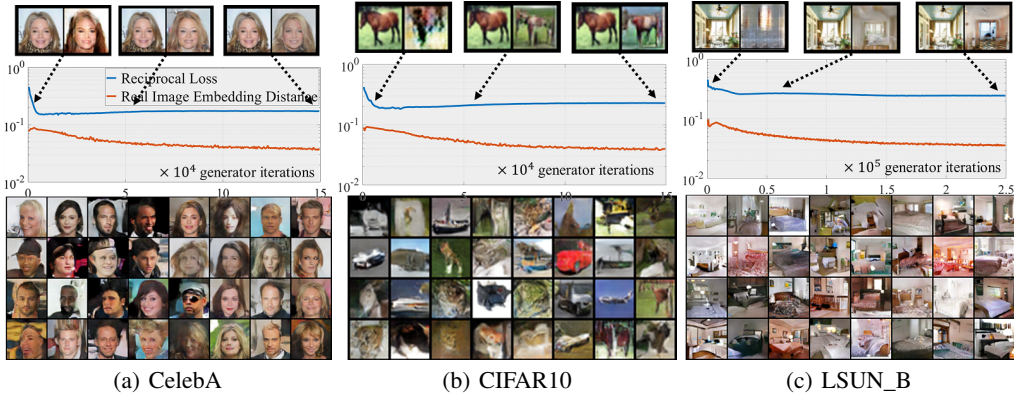

| (a) CelebA | (b) CIFAR10 | (c) LSUN_B |

Figure 3: The convergence curves and images generated by the proposed RCF-GAN from Gaussian noise, under the DCGAN [42] structure. Note that the curves were plotted by an average over a moving window, with $500$ iterations.

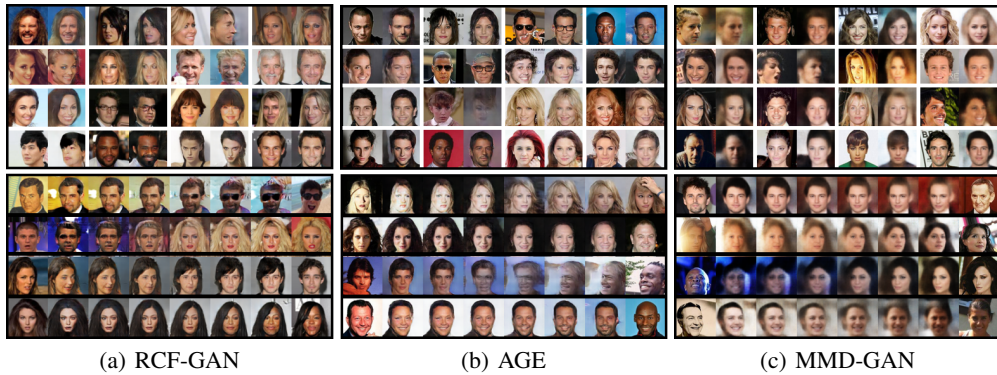

| (a) RCF-GAN | (b) AGE | (c) MMD-GAN |

Figure 4: Image reconstruction (upper panel) and interpolation (lower panel) by the proposed RCF-GAN, AGE [20] and MMD-GAN [13] in the CelebA dataset, under the DCGAN [42] structure. The upper panel shows the reconstructed images (in even columns) corresponding to the original images (in odd columns). The lower panel displays the linear interpolation in the embedded domain.

and reconstruction. This is beneficial because besides randomly generating real images, RCF-GAN is able to bi-directionally reconstruct and interpolate real images. In contrast, although MMD-GANs employ a third module to implement an auto-encoder, the decoded images are severely blurred.

Moreover, the proposed RCF-GAN subjectively achieved better reconstruction and interpolation than the AGE, by generating less blurred and more accurate images (for example, correct skin and hair colours). This is quantified in Table 1, which shows that the images reconstructed by our RCF-GAN are superior to those from the AGE. More importantly, by comparing with the FID and KID scores in Table 1, the images from the proposed RCF-GAN are consistently superior, whilst the quality of the reconstructed images in the AGE is significantly inferior to its random generated images. This also indicates the effectiveness of the unified structure of our RCF-GAN.

## 5  Conclusion

We have introduced an efficient generative adversarial net (GAN) structure that seamlessly combines the IPM-GANs and auto-encoders. In this way, the reciprocal in the proposed RCF-GAN ensures the equivalence between the embedded and data domains, whereas in the embedded domain the comparison of two distributions is strongly supported by the proposed powerful characteristic function (CF) loss, together with the physically meaningful phase and amplitude information, and an efficient sampling strategy. The reciprocal, accompanied with the proposed anchor design, has been shown to also stabilise the convergence of the adversarial learning in the proposed RCF-GAN, and at the same time to benefit from meaningful comparisons in the embedded domain. Consequently, the experimental results have demonstrated the superior performances of our RCF-GAN in both generating images and reconstructing images.

# 6 Broader Impact

A combination of the auto-encoder and GANs has been extensively studied, and has been shown to achieve a broader data generation and reconstruction. The RCF-GAN proposed in this paper provides a neat and new structure in the combination. The studies of GANs and those design on probabilistic auto-encoders basically start from different perspectives because the former serves for the generation, or it "decodes" from random noise, whilst the latter, as its name implies, focuses on encoding to summarise information. Although there are extensive attempts on combining those two structures, they typically embed one into the other as components such as by using an auto-encoder as a discriminator in GANs or using an adversarial idea in an auto-encoder. This paper provides a way of equally treating the two structures; the proposed structure, which contains only two modules, can be regarded both as an "encoder-decoder" and "discriminator-generator". The proposed combination benefits both, that is, it equips an auto-encoder the ability to meaningfully encode via matching in the embedded domain, whilst ensuring the convergence of the adversarial as a GAN.

Moreover, instead of being a component to measure the distance as in the W-GAN, regarding the *critic* as an independent feature mapping module with a sufficient distance metric is beneficial to allow learning in the embedded domain for any types of feature extraction models, such as the deep canonical correlation analysis net and graph auto encoder. A large amount of unsupervised learning models, then, can be connected and improved with the adversarial learning.

Another potential benefit of our work is to bring the general concept of the characteristic function (CF) into practice, by providing efficient sampling methods. The CF has been previously studied as a powerful tool in theoretical probabilistic analysis, while its practical applications have been limited due to complex functional forms. We should also highlight the physical meaning of the CF components introduced in this paper. It is a well known experimental phenomenon that the phase of discrete Fourier transform of images captures the saliency information, which motivates a large volume of works in saliency detection. This paper gives a probabilistic explanation to this, paving the way for future work to embark upon this intrinsic relationship.

## Acknowledgments and Disclosure of Funding

Shengxi Li wishes to thank Imperial Lee Family Scholarship for the support of his research.

## Footnotes

[2]To our best knowledge, the only exception is the AGE [20], which adopts two modules in an auto-encoder under a max-min problem and different losses. Please see the *Related Works* for the difference.

[3]A very recent independent work [27] named OCF-GAN also employs the CF as a replacement by using the same structure of MMD-GANs. The proposed RCF-GAN is substantially different from that in [27]. We refer to the *Related Works* in the supplementary material for a detailed explanation.

[4]This phenomenon has been discovered in the Fourier representation of signals [37, 38]. We validate that this also holds in probabilistic distributions.

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
