[Supplementary Material]

# Supplementary Material of the Paper: Reciprocal Adversarial Learning via Characteristic Functions

**Shengxi Li**[*]    **Zeyang Yu**    **Min Xiang**    **Danilo Mandic**
Imperial College London
{shengxi.li17, z.yu17, m.xiang13, d.mandic}@imperial.ac.uk

## 1 In-depth Analysis

### 1.1 Phase and Amplitude in the CF Loss

We report the effects of $\alpha$ on training the overall RCF-GAN in Fig. 1. From this figure, we can find that the proposed RCF-GAN is robust to the choice of $\alpha$, as when $\alpha$ ranges from 0.1 to 0.9, the RCF-GAN still achieved relatively superior generations. More importantly, we have not witnessed any mode collapse generations in all experiments. Although $\alpha = 0.5$ was a default and mainly used in our experiments, varying $\alpha$ could even achieve better performances. For example, for the dataset without complex and diversified scenarios (e.g., CelebA), imposing amplitude by $\alpha = 0.75$ increased the FID (KID) from 15.86 (0.011) to 13.84 (0.009). As the amplitude relates to the diversity measurement in the CF loss, the increment may come from enhancing the richness of generated faces. On the other hand, for some complicated scenarios (e.g., CIFAR-10), keeping the mean of data generation (that is, focusing on the phase) could be more beneficial (e.g., $\alpha = 0.1$).

Fig. 1 further shows an illustrative example on some over-weighted examples from CelebA. When over-weighting the phase ($\alpha = 0.1$), the generated images tend to be whitened and blurred, with their interpolation less smooth. This indicates that RCF-GAN tended to learn the average (mean) information of the data. On the contrary, when the amplitude was over-weighted ($\alpha = 0.9$), the generated images were over-saturated and with noisy artefacts, meaning that the RCF-GAN was likely to learn diversified content, even though some learnt faces were inaccurate. Therefore, the physical meaning of the proposed CF loss can provide a feasible way of understanding and evaluating generation details where the KID and FID metrics cannot reflect.

Figure 1: FID and KID scores for different $\alpha$, under the DCGAN [1] structure. Observe the embedded space (by interpolated images) of the proposed RCF-GAN, which was learnt with different $\alpha$ on CelebA dataset.

---

[*]Corresponding author

Table 1: Ablation study on the CelebA dataset. The proposed RCF-GAN was evaluated and compared to the one without (w/o) reciprocal requirement ($\lambda = 0$) and without anchor design.

| | FID | | | 
| --- | --- | --- | --- |
| | w/o reciprocal | w/o anchor | RCF-GAN |
| G. | 59.39±0.37 | 17.80±0.20 | 15.86±0.08 |
| R. | >100 | >100 | 14.82±0.12 |
| | KID | | |
| | w/o reciprocal | w/o anchor | RCF-GAN |
| G. | 0.046±0.001 | 0.010±0.000 | 0.011±0.001 |
| R. | >0.060 | >0.060 | 0.009±0.000 |

Note: G. is for image random generation and R. for image reconstruction.

## 1.2 Ablation Study

The roles of the two key distinguishing elements of the proposed RCG-GAN are now evaluated via an ablation study on the CelebA dataset. There are the term $\lambda$ that controls the reciprocal together with the anchor design. The results in Table 1 showed that without the reciprocal loss (by setting $\lambda = 0$) the overall generation was largely degraded and the reconstruction even completely failed. This, on the one hand, highlights the necessity of the reciprocal loss in our work; on the other hand, it also validates the correctness of the theoretical guarantee in Lemma 4, which is an important requirement that also motivates the auto-encoder structure in our RCF-GAN. Moreover, Table 1 also validates the effectiveness of the proposed anchor design. Without the anchor design, the generation still works, however, because the minimisation of $\mathcal{C}_{\mathcal{T}}(f(\overline{\mathcal{X}}), \mathcal{Z})$ by the anchor does no longer exist, the mapping of real images $f(\overline{\mathcal{X}})$ might not completely fall into the support of $\mathcal{Z}$, thus leading to poor reconstructions. Therefore, in order to successfully generate and reconstruct images, the reciprocal and anchor architecture are necessary in the proposed RCF-GAN.

Figure 2: The ResNet $128 \times 128$ structure adopted in this work. Note that the numbers in red colour represent the channel setting of ResNet $64 \times 64$.

## 2 Advancements under ResNet Structure

The scalability of the proposed RCF-GAN was further evaluated over complex net structures and higher image sizes. Specifically, we trained RCF-GAN under the ResNet structure, in terms of image sizes of $64 \times 64$ and $128 \times 128$. The ResNet structure under image size $64 \times 64$ was exactly the same as that in [4]. We extended this structure to the image size $128 \times 128$ in a similar way to the DCGAN, which is shown[2] in Fig. 2. We adopted the spectral normalisation instead of the layer normalisation in the ResNet experiments and also encourage to refer to our implementations for more detail.

The FID and KID scores are given in Table 2, and the results of randomly generating, reconstructing and interpolating $128 \times 128$ images are provided in Fig. 3. More results on image sizes of $64 \times 64$ can be found in Fig. 4.

Table 2: The FID and KID scores under the ResNet structure in terms of $64 \times 64$ and $128 \times 128$ image sizes. The result of Sphere GAN was obtained from the original article [2]. We ran the available code of the OCF-GAN-GP [3] with its implemented ResNet structure because it failed to converge in our structure given in Fig. 2.

| $64 \times 64$ | FID | | KID | |
| --- | --- | --- | --- | --- |
| | Celeba | LSUN_B | Celeba | LSUN_B |
| Sphere GAN | — | 16.9 [2] | —- | —- |
| RCF-GAN | 9.02±0.22 | 8.76±0.07 | 0.006±0.001 | 0.005±0.001 |
| RCF-GAN (R.) | 8.06±0.08 | 7.89±0.05 | 0.003±0.000 | 0.002±0.000 |
| $128 \times 128$ | | | | |
| OCF-GAN-GP | 20.78±0.15 | 21.82±0.20 | 0.015±0.001 | 0.014±0.001 |
| RCF-GAN | 10.71±0.11 | 10.32±0.13 | 0.006±0.000 | 0.005±0.001 |
| RCF-GAN (R.) | 13.01±0.15 | 8.64±0.10 | 0.006±0.000 | 0.003±0.000 |
| Note: R. is for image reconstruction. | | | | |

(a) Generation      (b) Reconstruction      (c) Interpolation

Figure 3: Random generation, reconstruction and interpolation by the proposed RCF-GAN by ResNet in terms of image size $128 \times 128$. The upper panel shows images of the CelebA dataset and the lower panel is for the LSUN_B dataset.

## 3 Proofs

We repeat the CF loss in the paper for the convenience of referring in the following proofs.

$$\mathcal{C}_{\mathcal{T}}(\boldsymbol{\mathcal{X}}, \boldsymbol{\mathcal{Y}}) = \int_{\mathbf{t}} \left( \underbrace{(\Phi_{\boldsymbol{\mathcal{X}}}(\mathbf{t}) - \Phi_{\boldsymbol{\mathcal{Y}}}(\mathbf{t}))(\Phi_{\boldsymbol{\mathcal{X}}}^*(\mathbf{t}) - \Phi_{\boldsymbol{\mathcal{Y}}}^*(\mathbf{t}))}_{c(\mathbf{t})} \right)^{1/2} dF_{\mathcal{T}}(\mathbf{t}), \tag{1}$$

**Lemma 1.** *The discrepancy between $\boldsymbol{\mathcal{X}}$ and $\boldsymbol{\mathcal{Y}}$, given by $\mathcal{C}_{\mathcal{T}}(\boldsymbol{\mathcal{X}}, \boldsymbol{\mathcal{Y}})$ in (1), is a distance metric when the support of $\mathcal{T}$ resides in $\mathbb{R}^m$.*

*Proof.* We here prove the non-negativity, symmetry and triangle properties that are required as a valid distance metric.

**Non-negativity:** Based on the definition of $\mathcal{C}_{\mathcal{T}}(\boldsymbol{\mathcal{X}}, \boldsymbol{\mathcal{Y}})$ in (1), the term $\mathcal{C}_{\mathcal{T}}(\boldsymbol{\mathcal{X}}, \boldsymbol{\mathcal{Y}})$ is non-negative because $c(\mathbf{t}) \geq 0$ for all $\mathbf{t}$. We next prove when the equality holds.

- $\boldsymbol{\mathcal{X}} =^d \boldsymbol{\mathcal{Y}} \to \mathcal{C}_{\mathcal{T}}(\boldsymbol{\mathcal{X}}, \boldsymbol{\mathcal{Y}}) = 0$: This is evident because $\Phi_{\boldsymbol{\mathcal{X}}}(\mathbf{t}) = \Phi_{\boldsymbol{\mathcal{Y}}}(\mathbf{t})$ for all $\mathbf{t}$.

(a) Generation　　　　　　　(b) Reconstruction　　　　　　　(c) Interpolation

Figure 4: LSUN Church images generated, reconstructed and interpolated by the RCF-GAN by ResNet in terms of image size $64 \times 64$. In this experiment, the employed ResNet was slightly different from the one in Fig. 2 by using the layer normalisation.

- $\mathcal{X} =^d \mathcal{Y} \leftarrow \mathcal{C}_{\mathcal{T}}(\mathcal{X}, \mathcal{Y}) = 0$: Given that the support of $\mathcal{T}$ is $\mathbb{R}^m$, $\int_{\mathbf{t}} \sqrt{c(\mathbf{t})} dF_{\mathcal{T}}(\mathbf{t}) = 0$ exists if and only if $c(\mathbf{t}) = 0$ everywhere. Therefore, $\Phi_{\mathcal{X}}(\mathbf{t}) = \Phi_{\mathcal{Y}}(\mathbf{t})$ for all $\mathbf{t} \in \mathbb{R}^m$. According to the *Uniqueness Theorem* of the CF, we have $\mathcal{X} =^d \mathcal{Y}$.

Therefore, $\mathcal{C}_{\mathcal{T}}(\mathcal{X}, \mathcal{Y}) \geq 0$, and the equality holds if and only if $\mathcal{X} =^d \mathcal{Y}$.

**Symmetry:** This is obvious for the symmetry of $c(\mathbf{t})$, thus yielding $\mathcal{C}_{\mathcal{T}}(\mathcal{X}, \mathcal{Y}) = \mathcal{C}_{\mathcal{T}}(\mathcal{Y}, \mathcal{X})$.

**Triangle:** Because the CFs $\Phi_{\mathcal{X}}(\mathbf{t})$ and $\Phi_{\mathcal{X}}(\mathbf{t})$ are the elements of the normed vector space, we have the following inequality (also known as the Minkowski inequality),

$$
\begin{aligned}
&\int_{\mathbf{t}} |\Phi_{\mathcal{X}}(\mathbf{t}) - \Phi_{\mathcal{Z}}(\mathbf{t}) + \Phi_{\mathcal{Z}}(\mathbf{t}) - \Phi_{\mathcal{Y}}(\mathbf{t})| dF_{\mathcal{T}}(\mathbf{t}) \\
&\leq \int_{\mathbf{t}} |\Phi_{\mathcal{X}}(\mathbf{t}) - \Phi_{\mathcal{Z}}(\mathbf{t})| dF_{\mathcal{T}}(\mathbf{t}) + \int_{\mathbf{t}} |\Phi_{\mathcal{Z}}(\mathbf{t}) - \Phi_{\mathcal{Y}}(\mathbf{t})| dF_{\mathcal{T}}(\mathbf{t}).
\end{aligned}
\tag{2}
$$

Therefore, the triangle property of $\mathcal{C}_{\mathcal{T}}(\mathcal{X}, \mathcal{Y})$ follows as

$$
\mathcal{C}_{\mathcal{T}}(\mathcal{X}, \mathcal{Y}) \leq \mathcal{C}_{\mathcal{T}}(\mathcal{X}, \mathcal{Z}) + \mathcal{C}_{\mathcal{T}}(\mathcal{Z}, \mathcal{Y}).
\tag{3}
$$

This means that $\mathcal{C}_{\mathcal{T}}(\mathcal{X}, \mathcal{Y})$ is a valid distance metric in measuring discrepancies between two random variables $\mathcal{X}$ and $\mathcal{Y}$.

This completes the proof.

**Lemma 2.** *If $\mathcal{X}$ and $\mathcal{Y}$ are supported on a finite interval $[-1, 1]^m$, $\mathcal{C}_{\mathcal{T}}(\mathcal{X}, \mathcal{Y})$ in (1) is still a distance metric for distinguishing $\mathcal{X}$ from $\mathcal{Y}$ for any $F_{\mathcal{T}}(\mathbf{t})$ that samples $\mathbf{t}$ within a small ball around $\mathbf{0}$.*

*Proof.* The proof of the triangle and symmetry properties is the same as those in Lemma 1. The non-negativity is also evident and the same as that in Lemma 1 but the equality holds for different conditions. We provide its proof in the following.

Before proceeding with the proof, we first quote Theorem 3 from Essen [6].

***Theorem 3 ([6])*** *The distributions of two random variables $\mathcal{X}$ and $\mathcal{Y}$ are the same when*

- $\Phi_{\mathcal{X}}(\mathbf{t}) = \Phi_{\mathcal{Y}}(\mathbf{t})$ *in an interval around $\mathbf{0}$;*

- $\beta_k = \int_x x^k dF_{\mathcal{X}}(x) < \infty$ *for $k = 0, 1, 2, 3, \ldots$*

- $\sum_{k=1}^{\infty} 1/\beta_{2k}^{1/2k}$ *diverges, which means that the moment problem of $\beta_k$ is determined and unique.*

It is the fact that only requiring $\Phi_{\mathcal{X}}(\mathbf{t}) = \Phi_{\mathcal{Y}}(\mathbf{t})$ in an interval around $\mathbf{0}$ does not ensure the equivalence between two distributions without any other constraints, also given the counterexample provided in [6]. This equivalence cannot be ensured even when all the moments are matched. The third condition, intuitively, guarantees this equivalence by restricting that the moment does not increase "extremely" fast when $k \to \infty$.

In Lemma 2 of this work, we bound $\mathcal{X}$ and $\mathcal{Y}$ by $[-1, 1]$, thus having $|\beta_k| \leq 1 < \infty$ and $1/\beta_{2k}^{1/2k} \geq 1$ so that $\sum_{k=1}^{\infty} 1/\beta_{2k}^{1/2k}$ diverges. In this case, according to Theorem 3, we have $\Phi_{\mathcal{X}}(\mathbf{t}) = \Phi_{\mathcal{Y}}(\mathbf{t})$ when $\mathbf{t}$ samples around $\mathbf{0} \to \mathcal{X} =^d \mathcal{Y}$. Conversely, it is obvious that $\mathcal{X} =^d \mathcal{Y} \to \Phi_{\mathcal{X}}(\mathbf{t}) = \Phi_{\mathcal{Y}}(\mathbf{t})$ for all $\mathbf{t}$. Therefore, as for bounded $\mathcal{X}$ and $\mathcal{Y}$, sampling around $\mathbf{0}$ is sufficient to ensure the symmetry, triangle, non-negativity (together with the uniqueness when the equality holds) properties of $\mathcal{C}_{\mathcal{T}}(\mathcal{X}, \mathcal{Y})$.

This completes the proof.

**Lemma 3.** *The metric $\mathcal{C}_{\mathcal{T}}(\mathcal{X}, \mathcal{Y})$ is bounded and differentiable almost everywhere.*

*Proof.* We first show the boundedness of $\mathcal{C}_{\mathcal{T}}(\mathcal{X}, \mathcal{Y})$ by observing

$$
\begin{aligned}
0 \leq \mathcal{C}_{\mathcal{T}}(\mathcal{X}, \mathcal{Y}) &= \int_{\mathbf{t}} |\Phi_{\mathcal{X}}(t) - \Phi_{\mathcal{Y}}(t)| dF_{\mathcal{T}}(\mathbf{t}) \\
&\leq \int_{\mathbf{t}} |\Phi_{\mathcal{X}}(t)| dF_{\mathcal{T}}(\mathbf{t}) + \int_{\mathbf{t}} |\Phi_{\mathcal{Y}}(t)| dF_{\mathcal{T}}(\mathbf{t}) \leq 1 + 1 = 2,
\end{aligned}
\tag{4}
$$

where the second inequality is obtained via the Minkowski inequality and the third one by the fact that the maximal modulus of the CF is $1$. It should be pointed out that this property is important and advantageous because in this way our cost is bounded automatically. Otherwise, we may need to bound $f \in \mathcal{F}$ to ensure an existence of the supremum of some IPMs (such as the dual form of the Wasserstein distance used in the W-GAN).

To prove the differentiable property, we first expand $c(\mathbf{t})$ in $\mathcal{C}_{\mathcal{T}}(\mathcal{X}, \mathcal{Y}) = \int_{\mathbf{t}} \sqrt{c(\mathbf{t})} dF_{\mathcal{T}}(\mathbf{t})$ as

$$
\begin{aligned}
c(\mathbf{t}) = (\text{Re}\{\Phi_{\mathcal{X}}(\mathbf{t})\} &- \text{Re}\{\Phi_{\mathcal{Y}}(\mathbf{t})\})^2 \\
&- (\text{Im}\{\Phi_{\mathcal{X}}(\mathbf{t})\} - \text{Im}\{\Phi_{\mathcal{Y}}(\mathbf{t})\})^2,
\end{aligned}
\tag{5}
$$

where $\text{Re}\{\Phi_{\mathcal{X}}(\mathbf{t})\} = \mathbb{E}_{\mathcal{X}}[\cos(\mathbf{t}^T \mathbf{x})]$ denotes the real part of the CF and $\text{Im}\{\Phi_{\mathcal{X}}(\mathbf{t})\} = \mathbb{E}_{\mathcal{X}}[\sin(\mathbf{t}^T \mathbf{x})]$ for its imaginary part. Therefore, by regarding $c(\mathbf{t})$ as a mapping $\mathbb{R}^m \to \mathbb{R}$, it is differentiable almost everywhere[3].

This completes the proof.

**Lemma 4.** *Denote the distribution mapping by $\overline{\mathcal{Y}} =^d g(\mathcal{Z})$. Given two functions $f(\cdot)$ and $g(\cdot)$ that are reciprocal on the supports of $\overline{\mathcal{Y}}$ and $\mathcal{Z}$, that is, $\mathbb{E}_{\mathcal{Z}}[||\mathbf{z} - f(g(\mathbf{z}))||_2^2] = 0$, we also have $\mathbb{E}_{\overline{\mathcal{Y}}}[||\overline{\mathbf{y}} - g(f(\overline{\mathbf{y}}))||_2^2] = 0$. More importantly, this yields the following equivalences: $\mathcal{C}_{\mathcal{T}}(f(\overline{\mathcal{X}}), f(\overline{\mathcal{Y}})) = 0 \Leftrightarrow \mathcal{C}_{\mathcal{T}}(\overline{\mathcal{X}}, \overline{\mathcal{Y}}) = 0 \Leftrightarrow \mathcal{C}_{\mathcal{T}}(f(\overline{\mathcal{Y}}), \mathcal{Z}) = 0$ and $\mathcal{C}_{\mathcal{T}}(f(\overline{\mathcal{X}}), \mathcal{Z}) = 0$.*

*Proof.* Because $\mathbb{E}_{\mathcal{Z}}[||\mathbf{z} - f(g(\mathbf{z}))||_2^2] = 0$ and $||\mathbf{z} - f(g(\mathbf{z}))||_2^2 \geq 0$, we have $\mathbf{z} = f(g(\mathbf{z}))$ for any $\mathbf{z}$ and $g(\mathbf{z})$ under the supports of $\mathcal{Z}$ and $\overline{\mathcal{Y}}$, respectively. We can obtain $g(\mathbf{z}) = g(f(g(\mathbf{z})))$ under the supports of $\mathcal{Z}$ and $\overline{\mathcal{Y}}$ as well; given that $\overline{\mathbf{y}} = g(\mathbf{z})$ by the definition, this results in $\overline{\mathbf{y}} = g(f(\overline{\mathbf{y}}))$. Then, we have $\mathbb{E}_{\overline{\mathcal{Y}}}[||\overline{\mathbf{y}} - g(f(\overline{\mathbf{y}}))||_2^2] = 0$. Therefore, the function $g(\cdot)$ is a unique inverse of the function $f(\cdot)$, and vice versa, which also indicates that the two functions are bijective.

The bijection of the function $f(\cdot)$ possesses many desirable properties between the domains of $\overline{\mathcal{Y}}$ and $\mathcal{Z}$, thus ensuring the equivalences between their CFs. Specifically, without loss of generality, we assume $\mathcal{C}_{\mathcal{T}}(\overline{\mathcal{X}}, \overline{\mathcal{Y}}) = 0$, which means

$$
\int_{\overline{\mathbf{x}}} e^{j\overline{\mathbf{t}}^T \overline{\mathbf{x}}} dF_{\overline{\mathcal{X}}}(\overline{\mathbf{x}}) = \int_{\overline{\mathbf{y}}} e^{j\overline{\mathbf{t}}^T \overline{\mathbf{y}}} dF_{\overline{\mathcal{Y}}}(\overline{\mathbf{y}}), \quad \text{for all } \overline{\mathbf{t}}.
\tag{6}
$$

Then, given the bijection $f(\cdot)$ by $\mathcal{X} = f(\overline{\mathcal{X}})$ and $\mathcal{Y} = f(\overline{\mathcal{Y}})$, we obtain $\mathbf{x} = f(\overline{\mathbf{x}}) = f(\overline{\mathbf{y}}) = \mathbf{y} \Leftrightarrow \overline{\mathbf{x}} = \overline{\mathbf{y}}$, for any realisations $\mathbf{x}$ and $\mathbf{y}$ from $\mathcal{X}$ and $\mathcal{Y}$. We then have the following equivalence between the CFs of $\mathcal{X} = f(\overline{\mathcal{X}})$ and $\mathcal{Y} = f(\overline{\mathcal{Y}})$,

$$\int_{\overline{\mathbf{x}}} e^{j\overline{\mathbf{t}}^T \overline{\mathbf{x}}} dF_{\overline{\mathcal{X}}}(\overline{\mathbf{x}}) = \int_{\overline{\mathbf{y}}} e^{j\overline{\mathbf{t}}^T \overline{\mathbf{y}}} dF_{\overline{\mathcal{Y}}}(\overline{\mathbf{y}}), \quad \text{for all } \overline{\mathbf{t}}$$

$$\Leftrightarrow \int_{\overline{\mathbf{x}}} e^{j\mathbf{t}^T f(\overline{\mathbf{x}})} dF_{\overline{\mathcal{X}}}(\overline{\mathbf{x}}) = \int_{\overline{\mathbf{y}}} e^{j\mathbf{t}^T f(\overline{\mathbf{y}})} dF_{\overline{\mathcal{Y}}}(\overline{\mathbf{y}}), \quad \text{for all } \mathbf{t} \tag{7}$$

$$\Leftrightarrow \int_{\mathbf{x}} e^{j\mathbf{t}^T \mathbf{x}} dF_{\mathcal{X}}(\mathbf{x}) = \int_{\mathbf{y}} e^{j\mathbf{t}^T \mathbf{y}} dF_{\mathcal{Y}}(\mathbf{y}), \quad \text{for all } \mathbf{t}.$$

Therefore, we have $\mathcal{C}_{\mathcal{T}}(f(\overline{\mathcal{X}}), f(\overline{\mathcal{Y}})) = 0$. Furthermore, we also have $f(\overline{\mathcal{Y}}) =^d \mathcal{Z}$ due to $\mathbb{E}_{\mathcal{Z}}[\|\mathbf{z} - f(g(\mathbf{z}))\|_2^2] = 0$. Therefore, we have the following equivalences: $\mathcal{C}_{\mathcal{T}}(\overline{\mathcal{X}}, \overline{\mathcal{Y}}) = 0 \Leftrightarrow \mathcal{C}_{\mathcal{T}}(f(\overline{\mathcal{X}}), f(\overline{\mathcal{Y}})) = 0 \Leftrightarrow \mathcal{C}_{\mathcal{T}}(f(\overline{\mathcal{Y}}), \mathcal{Z}) = 0$ and $\mathcal{C}_{\mathcal{T}}(f(\overline{\mathcal{X}}), \mathcal{Z}) = 0$.

This completes the proof.

## 4 Related Works

**IPM-GANs:** Instead of the naive weight clipping in the W-GAN [9], the gradient penalty in W-GAN (W-GAN-GP) was proposed to mitigate the heavily constrained *critic* by penalising the gradient norm [4], followed by a further elegant treatment by restricting the largest singular value of the net weights [5]. It has been understood that although the *critic* cannot search within all satisfied Lipschitz functions [10, 11], the *critic* still performs as a way of transforming high dimensional but insufficiently supported data distributions into low dimensional yet broadly supported (simple) distributions in the embedded domain [12]. Comparing the embedded statistics, however, is much easier. For example, Cramer GAN compares the mean with an advanced $\mathcal{F}$ from the Cramer distance to correct the biased gradient [13], whilst McGAN [14] explicitly compares the mean and the covariance in the embedded domain. Fisher GAN employs a scale-free Mahalanobis distance and thus a data dependent $\mathcal{F}$ [15], which is basically the Fisher-Rao distance in the embedded domain between two Gaussian distributions with the same covariance. The recent Sphere GAN further compares higher-order moments up to a specified order, and avoids the Lipschitz condition by projecting onto a spherical surface [2]. Moreover, in a non-parametric way, BE-GAN directly employs an auto-encoder as the *critic*, whereby the auto-encoder loss was compared through embedded distributions [16]. The sliced Wasserstein distance has also been utilised into measure the discrepancy in the embedded domain [17]. Another non-parametric metric was achieved by the kernel trick of the MMD-GAN [18, 12], which treats $\mathcal{F}$ as the reproducing kernel Hilbert space. However, one of the most powerful ways of representing a distribution, the CF, is still to be fully explored. More importantly, our RCF-GAN both directly compares the embedded distributions and also potentially generalises the MMD-GAN by flexible sampling priors.

Moreover, a very recent independent work [3] named OCF-GAN also employs the CF as a replacement by using the same structure of MMD-GANs. Our RCF-GAN is substantially different from that in [3]:

• Our *critic* operates as semantic embeddings and learns a meaningful embedded space, instead of being a component to build complete metrics as the existing GANs (e.g., OCF-GAN, MMD-GANs and W-GANs) do.

• Our CF design, is novel in its triangle anchor design with $l_1$-norm (to stabilise convergence), meaningful analysis of amplitude and phase (to favour other distribution alignment tasks), $t$-net of outputting scales (to automatically optimise $\mathcal{T}$ distribution types), and useful supporting theory (to correctly and efficiently use CF in practice).

• Our RCF-GAN seamlessly combines the auto-encoder and GANs by using only two neat modules while achieving the state-of-the-art performances, whereas the majority of adversarial learning structures use at least three modules with auto-encoding separated from GANs.

Consequently, the results in [3] were reported given all images rescaled to the size of $32 \times 32$, while our RCF-GAN consistently outperforms in various ways including high resolutions, net structures and functionalities.

**Auto-encoders in an adversarial way:** To address the smoothing artefact of the variational auto-encoder [19], several works aim to incorporate the adversarial style in (variational) auto-encoders, in the hope of gaining clear images whilst maintaining the ability of reconstruction. These mostly consist of at least three modules, an encoder, a decoder, and an adversarial modules [20–25]. To the best of our knowledge, there is one exception, called the adversarial generator encoder (AGE) [26], which incorporates two modules in adversarially training an auto-encoder under a *max-min* problem. The AGE still assumes the Gaussianity in the embedded distributions and only compares the mean and the diagonal covariance matrix; this is basically insufficient in identifying two distributions, and requires the pixel domain loss to be utilised supplementally in implementations. Our work, stilling playing a min-max problem, is fundamentally different from the AGE, as the auto-encoder in our RCF-GAN is a necessity to achieve the theoretical guarantee of a reciprocal, with the proposed anchor design. In contrast, without the auto-encoder, the AGE could still work by its Theorems 1 and 2 [26]. Furthermore, other than the first- and second-order moments, our work fully compares the discrepancies in the embedded domain via CFs. Benefiting from the powerful non-parametric metric via the CFs, our RCF-GAN only adversarially learns distributions in the embedded domain, that is, on a semantically meaningful manifold, without the need of any operation on the data domain.

## Footnotes

[2]Please note that the *critic* of our ResNet $128 \times 128$ structure is slightly different from that in the spectral GAN [5]. We adopted a symmetric (mirror) structure of the generator, whereby the spectral GAN used an asymmetric one. Although RCF-GAN still works under the structure of the spectral GAN, we believe that the mirror structure can well reflect the proposed reciprocal idea and is also a natural extension of the ResNet $64 \times 64$ in [4]. The parameter size in our ResNet structure is slightly smaller than that in the spectral GAN.

[3] We note that $c(\mathbf{t})$ is not necessarily complex differentiable because it does not satisfy the Cauchy-Riemann equations. It is the fact that *nonconstant purely real-valued functions* are not complex differentiable because their Cauchy-Riemann equations are not satisfied. However, in our case, it is differentiable as it is regarded as mappings in the real domain. Please refer to [7, 8] for more detail in the $\mathbb{CR}$ calculus.