[Reviews · NeurIPS 2020]

Review 1

Summary and Contributions: The paper introduces a generalization for IPM-GANs by comparing distributions through their characteristic functions (CF). Relying on the theoretical guarantees involving uniqueness of CF and weak convergence of empirical CF, the authors devise a Reciprocal CF-GAN uniting auto-encoders and GANs. The paper demonstrates their empirical effectiveness in both image generation and reconstruction tasks. More broadly, the paper suggests a new approach to compare distributions in carefully embedded domains that is a promising direction in machine learning. === Post rebuttal and discussion update === I thank the authors and reviewers for the rebuttal and the discussion, I tend to keep my score.

Strengths: The underlying idea is a novel and significant contribution to the field of generative modelling, to my knowledge. It has an excellent theoretical grounding in treating the distance between distributions with a powerful tool of characteristic functions and a proper empirical evaluation demonstrating its superiority over previous approaches. The paper may also bring more future work utilizing empirical CFs in other fields of machine learning.

Weaknesses: Not necessarily a weakness, but rather a question for further investigation. The authors use CF of a unimodal distribution, and it is interesting to study if it is limiting expressivity of the model for more complex datasets than MNIST. Also, should not t-net be treated as a discriminator in the embedded domain, while for f_theta it will make more sense to be noted as just an encoder? It seems that CF and the t-net are the parts that actually implement critic's functionality.

Correctness: Yes, claims, method, and empirical methodology seem good.

Clarity: The paper content is undoubtedly dense, given this the paper is well-written, it is hard to imagine a better exposition. It should be mentioned that while related work is postponed to the supplementary materials, the introduction creates a comprehensible context.

Relation to Prior Work: As mentioned above, while the related work is well discussed in the introduction, the more detailed exposition (Related Work section) can only be found in the Appendix.

Reproducibility: Yes

Additional Feedback:


Review 2

Summary and Contributions: Authors propose a novel statistical distance C_T(X, Y) parameterized by a distribution T. The proposed metric equals the expected squared difference between values of characteristic functions of compared random measures evaluated at locations sampled from T. Authors show that C_T is indeed a metric, and that if we compare distributions with finite supports, T can be restricted to distribution supported in a ball around the origin. Authors suggest to compare empirical estimates of CFs, and to parameterize T as a scale mixture of normals with scale parameters generated by a neural network. Authors show that if Y = g(Z), and f is an inverse of g, then C_T(X, Y) = 0 iff C_T(f(X), f(Y)) = 0 and therefore propose the final algorithm that 1) trains f and T to minimize the discrepancy between f(X) and a Gaussian, f(g(Z)) and a Gaussian, while keeping f equal to the inverse of g, while 2) training g to minimize the distance between f(X) and f(g(Z)). Authors show that the resulting training procedure yields better FID and KID scores on CIFAR, CelebA and LSUN then DCGAN W-GAN and MMD-GAN, and discuss how different components of the loss might be interpreted.

Strengths: Authors clearly introduce a novel technique for distribution alignment, providing solid theoretical results to ground their claims, as well as solid empirical evaluation. The way authors employ characteristic functions in this paper is novel and might spark new ideas in the distribution alignment community. The paper is well-written and easy to follow, the introduction of CF for distribution alignment contains good examples and illustrations.

Weaknesses: My primary concern is that: 0. The paper seems to propose two ideas: 1) measuring distance between distributions as an expected squared difference between empirical characteristic functions evaluated at points sampled according to some adversarially learned distribution T; 2) the reciprocal training of adversarial autoencoders, i.e. adversarially aligning embeddings of X and Y, while making sure that these embeddings follow the Gaussian distribution and minimize the reconstruction loss. I wonder whether the impact of these two design choices can be evaluated independently: 1) seeing how direct minimization of C_T(X, g(Z)) wrt g performs compared to the model with a dedicated encoder/critic; 2) replacing C_T in Algorithm 1 with MMD / Sliced Wasserstein Distance or another statistical distance (moreover, distance to a Gaussian can often be estimated in closed form); does Lemma 4 hold for other statistical distances? And there are some things that I must have misunderstood. 1. In general, authors discuss in great details possible interpretations of phase and amplitude components of CFs, but cram a lot of content critical to proper understanding of the final model on the first half of page 6. For example, in lines 214-215: “we further re-design the critic loss by finding an anchor as C(f(Y),Z) − C(f(X),Z)” - it is still not clear to me what “anchors” authors are referring to. And, consequently. I did not quite understand why authors minimize the reconstruction loss on generated images only (the lambda * (..) term in the critic loss), and why they train f(X) to be as far from a Gaussian as possible, if Lemma 4 talks about low C_T(f(X), Z). 2. Proposition 1 claims that the maximum of C_T over possible choices of T is attained when T is a point mass placed where characteristic functions differ the most. Authors claim that this degenerate solution is undesirable, but it is not clear to me why. This would let you get rid of T and define C(X, Y) as just an l-infinity distance between empirical CFs. What is so bad about it? 3. In Lemma 4, if I understand it correctly, it seems odd that if Y = f(Z) and f is an inverse of g, then C_T(f(Y), Z) = 0 <=> C_T(X, Y) = 0, because the former C_T(f(Y), Z) = 0 does not include X and should always hold under the assumptions of this lemma. Maybe authors meant [ C(f(X), f(Y)) = 0 and C(f(X), Z) = 0 and C(f(Y), Z) = 0 ] <=> C(X,Y) = 0? Also, the statement itself seems odd to me: consider Z ~ Gaussian, X = (Z, 1), Y = g(Z) = (Z, -1). f(a, b) = a. f and g are reciprocal, and C_T(f(X), f(Y)) = 0, but C_T(X, Y) != 0. I am probably missing something. 4. It is not immediately apparent what is the difference between (\bar X, \bar Y) introduced on line 186 and (X, Y) used prior to that point. Bar variables are finite-sample distributions, whereas non-bar are random variables? Then It makes sense to connect them with empirical CFs introduced in lines 95-97. 8. Lines 140: “On the other hand, when training the amplitude only, we can obtain 140 diversified but inaccurate images (some digits are wrongly generated).” - these images actually look good, so it is not immediately obvious how exactly modeling the phase component would help.

Correctness: Apart from several statements I likely misunderstood (see Weaknesses), the claims in the paper seem correct.

Clarity: Apart from the first half of the page 6 9see weaknesses), the presentation is fluid.

Relation to Prior Work: Yes.

Reproducibility: Yes

Additional Feedback: This is a good submission. I am ready to change my vote to "accept" if either other reviewers find weaknesses I outlined invalid, or if authors clarify these points to me. Post-rebuttal: Most of my concerns were addressed in the rebuttal. It it difficult to explain an entire section, given space limitations, but from author's response to Q0-1, I conclude that authors will be able to make description on the first half of page 6 easier to comprehend. Authors seem to have ran experiments I outlined in Q0 but did not include them in the submission, so, conditioned on including these results into the final version, I increase my rating to "accept".


Review 3

Summary and Contributions: This paper investigates variants of Integral Probability Metric (IPM) that serve as the loss function to train implicit generative models (IGMs). In particular, the author considers the function class in IPM to be characteristic functions (CF) of distributions, of which the fundamental studies can be dated back to [1,2]. To enhance the discriminative power of the CF loss, the author optimizes the inverse Fourier transform distribution (equivalent to the kernel spectral distributions when the kernel is shift-invariant) by the neural network parametrization, which is also not new as first studied in [3,4]. The remaining novelty seems limited, which is about tweaking the critic loss into contrastive forms (i.e., the so-called reciprocal loss in this paper), plus an auto-encoder reconstruction loss. While the presented empirical results seem encouraging at first glance, the author missed an important comparison to the very-similar work [3,4]. Also, the author should carefully study the ablation study of (1) original CF critic loss without anchoring versus the proposed reciprocal form; and (2) the advantage of adding autoencoder reconstruction loss (lambda > 0 v.s. lambda=0). [1] Hilbert Space Embeddings and Metrics on Probability Measures, JMLR 2010 [2] Fast Two-Sample Testing with Analytic Representations of Probability Measures, NIPS 2015 [3] Implicit Kernel Learning, AISTATS 2019 [4] A Characteristic Function Approach to Deep Implicit Generative Modeling, CVPR 2020

Strengths: The empirical results seem promising

Weaknesses: (1) Limited novelty as using CF loss to train IGMs and optimize its Fourier distribution are both studied in [3,4]. (2) Lack ablation study to justify the advantage of reciprocal loss and reconstruction loss when training the critic.

Correctness: (1) In [3,4], they assume the critic function f to be Lipschitz, so that the optimized critics loss enjoys the weak topology, as well as being continuous everywhere and almost differentiable everywhere in the generator parameter theta. It somehow surprising and hard for me to believe Algorithm 1 can work without imposing such Lipschitz continuity on the critic function f. (2) Table 1 seems inconsistent with Table 1 of [4] and Table 2 of [3], especially the MMD-GAN results on the CIFAR-10 and CelebA datasets. Please elaborate.

Clarity: writing is clear in general. Please specify the difference of t_norm and t_net in Table 1.

Relation to Prior Work: The author did not discuss and compare to [3,4], which is very relevant works that also optimize/learn the inverse Fourier distribution for the IPM.

Reproducibility: Yes

Additional Feedback: The motivation of using reciprocal loss (i.e., the third line within the While loop of Algorithm 1) is not well justified, both theoretically and empirically. This issue limited the technical soundness of this paper.


Review 4

Summary and Contributions: This paper proposes a new GAN training method based on the CFs. The authors present an IPM-based approach to match the distributions of real/fake embeddings, which allows them to develop the GAN in an auto-encoder way. They refer to this efficient structure as the reciprocal CF GAN (RCF-GAN). The authors argue that RCF-GAN has several advantages compared to IPM-GAN: 1. RCF-GAN ensures the equivalence between the embedded and data domains 2. The new loss function based on CF has a clear physical meaning for the phase and amplitude. 3. The generation quality can be significantly improved. They compare the performance of RCF-GAN and IPM-GAN on several tasks, and the results show that the RCF-GAN performs better than IPM-GAN in terms of both generation and reconstruction. The advantage of the RCF-GAN lies in the clear physical meaning of the loss function and the simplicity of the training.

Strengths: This method is more mathematically grounded, the reciprocal nature of the loss function is the key to this method. The advantage of the RCF-GAN lies in the clear physical meaning when dealing with phase and amplitude information.

Weaknesses: The results show that RCF-GAN performs better than IPM-GAN, however it's a bit vague about whether the comparisons are based on comparable total number of model parameters. The only quantitative comparison is about the generation quality, so it's hard to judge the effectiveness of the reconstruction(interpolation) power of proposed method with the given evidence from the authors. Also, the authors claim that the loss function based has a clear physical meaning for the phase and amplitude, it would be better if they provide some explanations/analysis with the provided results.

Correctness: Correct

Clarity: The overall writing is clear, and the paper is easy to read.

Relation to Prior Work: Yes

Reproducibility: No

Additional Feedback: I think this method is promising. I have two comments: 1. It would be better to present more concrete evidence that the proposed method works well in terms of the generation quality and reconstruction. 2. It would be better to include more dataset to show that the method is scalable. I have read the author response and other reviewers' comments. I think the authors successfully addressed many raised concerns. I thereby increase my overall score.

[Author Response · NeurIPS 2020]

**General comments:** We thank all the reviewers for their insightful comments, and their unanimous positive comments
on the superior performance and high quality of our work. Our novelty has also been affirmed by R1, R2 and R4. We
believe that R3 might have misunderstood our work to be an extension of MMD-GANs (e.g., Refs [3,4] suggested by
R3). However, we should clarify that (1) our work differs completely from MMD-GANs, and (2) although Ref [4]
remains unpublished before the submission deadline, our work also achieves superior performances over Refs [3,4]. We
are confident that our work is novel because *our critic operates as semantic embeddings and learns a fruitful latent*
*space*, instead of being a component to build complete metrics as the existing GANs (e.g., MMD-GANs and W-GANs)
do. Theoretically, this avoids the Lipschitz constraint but it requires a reciprocal between the generator and critic, where
the *auto-encoder is a necessity* instead of just a plug-in. Our CF design (even given the unpublished work), is novel in
its ● *triangle anchor design with $l_1$-norm* (to stabilise convergence), ● *meaningful analysis of amplitude and phase* (to
favour other distribution alignment tasks), ● *t-net of outputting scales* (to optimise $\mathcal{T}$ distribution types), and ● *useful*
*theory*, e.g., Prop. 1 for ill-posed optimisation (a potential problem in Ref [3]) and Lemma 2 for reducing sets. Thus,
our RCF-GAN *seamlessly combines the auto-encoder and GANs by using only two neat modules while achieving s.o.t.a.*
*generation and reconstruction*, whereas MMD-GANs (and many others) typically use three modules, but reconstruct
and interpolate blurred images. Our supplementary material includes the *s.o.t.a.* results under adv-DCGAN and ResNet
structures on more datasets. Below we discuss the reviewers' comments and will address all of them in the revision.

**R1:** **[Expressivity]**: This is indeed an interesting topic. Although having proved the effectiveness in our RCF-GAN,
the unimodal setting may limit the expressivity on complex datasets and a worthy investigation could be using mixture
models (or learning models) to further improve the performance. **[$t$-net]**: Yes, exactly. In our code (to be released upon
acceptance) we put both $f(\cdot)$ and $t$-net in the critic, as they are optimised simultaneously. As $t$-net is optional (we can
directly use Gaussian samples), we separate it from $f(\cdot)$ but it belongs to the critic; this will be clarified in the revision.

**R2:** **[Q0]**: (1) Fig.2-b shows some results on MNIST by directly training $g(\mathbf{z})$. For other datasets, solely training $g(\mathbf{z})$
via the CF typically performs inferior, which in our preliminary trials on CelebA, obtained a 165 FID score (rough faces).
This also verifies the benefit of latent space comparison via our critic. Tailored $t$-net could be of help and we will provide
more results in the revision. (2) Yes, other metrics can be seamlessly adopted in our framework. However, we found that
our CF loss works better in practice because our $t$-net automatically optimises $\mathcal{T}$ distribution types. Our CF loss also
allows for more complex distributions (e.g., mixture models) to further improve model expressibility. **[Q1]**: (1) The
anchor refers to $\mathcal{Z}$ and the critic loss (minimising) is $-(C_{\mathcal{T}}(f(\overline{\mathcal{Y}}),\mathcal{Z})-C_{\mathcal{T}}(f(\overline{\mathcal{X}}),\mathcal{Z}))$. In the dynamic training process,

$\mathcal{Z}$ provides a pivot for the critic; please see the figure. This way, the critic can quickly map real data
$\overline{\mathcal{X}}$ to the support of $\mathcal{Z}$ via ⓑ (as Lemma 4 requires). (2) Lemma 4 ensures $\mathbb{E}_{\mathcal{Z}}[||\mathbf{z}-f(g(\mathbf{z}))||_2^2]=$

$0 \leftrightarrow \mathbb{E}_{\overline{\mathbf{y}}}[||\overline{\mathbf{y}}-f(g(\overline{\mathbf{y}}))||_2^2]=0$, i.e., the equivalence between reconstructing in the latent space (on
generated images) and the pixel domain (on real images). (3) As GANs, we adversarially train
$f(\overline{\mathcal{Y}})$ away from $\mathcal{Z}$ by ⓒ. **[Q2]**: $C_{\mathcal{T}}(\cdot,\cdot)$ is then not a valid metric for the degenerated $\mathcal{T}$ since
its support is not $\mathbb{R}^m$ as Lemma 1 requires (may have $C_{\mathcal{T}}(\mathcal{X},\mathcal{Y})=0$ but $\mathcal{X} \neq \mathcal{Y}$). **[Q3]**: We apologise for the typos as
pointed out by the reviewer. For the second question, the reviewer might have missed the need of identical supports on
$f$ and $g$. If $\mathcal{Z} \in \mathbb{R}^m$, the support of $\mathcal{Y}=(\mathcal{Z},-1)$ on $g$ is $\mathbb{R}^m$, which does not equal to the support of $(\mathcal{Z},b)$ on $f$ (i.e.,
$\mathbb{R}^{m+1}$), leading to $g(f(\mathcal{Z},1)) \neq (\mathcal{Z},1)$. **[Q4]**: $(\mathcal{X},\mathcal{Y})$ are for the random variables in the latent space and $(\overline{\mathcal{X}},\overline{\mathcal{Y}})$ for
the pixel domain. **[Q8]**: In Fig.2-b, using the amplitude only results in generating "wrong" numbers ("1" for digit 4, "6"
for digit 5), uneven characters, disconnected artefacts, etc, all of which can be relieved by adding the phase in training.

**R3:** **[Novelty]**: Please refer to lines 3-17 of this rebuttal. **[Ablation and anchor design]**: During the rebuttal period
we did the ablation study on CelebA and report the FIDs for the original (15.9), $\lambda=0$ (59.3) and without anchor (17.5).
Thus, ● without reconstruction loss, the generation largely degrades, *validating the correctness and necessity of our*
*reciprocal theory*; ● our motivation of anchoring is to find a pivot (i.e., $\mathcal{Z}$) during dynamic training process, where our
critic can quickly map real data to the support of $\mathcal{Z}$, as Lemma 4 requires. This is evidenced by the 1.6 FID gain. We
will provide more results in the revision. **[Lipschitz]**: Similar to other GANs (e.g., Fisher GAN and Sphere GAN), the
Lipschitz constraint is not a necessity in our RCF-GAN. Please refer to our proof. **[Consistency]**: The inconsistency on
CelebA with [4] is because [4] rescaled images in ALL datasets to a small size $32 \times 32$. For CIFAR10, our result is
consistent with [4], whereby both our and [4] are inconsistent with [3] (Table 2). Compared to [3,4], our RCF-GAN
also achieved the best performances. **[Notation]**: $t_{norm}$ uses fixed Gaussian and $t_{net}$ learns the $t$-net for ablation study.

**R4:** **[Model parameters]**: Our critic and generator are evaluated under the almost SAME number of model parameters
as W-GANs, whereas MMD-GANs need an extra decoder net. The only extra cost in our optional $t$-net is negligible
because it is a 3-layer FC net with the dimension of each layer less than 128. **[Reconstruction and datasets]**:
Fig.4 in the paper shows the image reconstruction and interpolation, validating our superior performances on *clear*
*reconstructions and semantic interpolations*. Our supplementary material further validates our scalable and consistent
superior performances on two extra structures (adv-DCGAN and ResNet) and one extra dataset (LSUN Church). We
will also release our code upon acceptance. **[Phase and amplitude]**: As the amplitude weight measures the model
diversity, the mode collapse can be efficiently relieved in our results. We will elaborate more upon this in the revision.

[Meta-Review · NeurIPS 2020]

The reviewers have reached a consensus that this work constitutes a novel and interesting extension of previous work on learning implicit models using GANs and integral probability metrics. The problems identified in the first round of reviewing were largely addressed in the author response, and I therefore can comfortably recommend accepting this paper.